# Ensemble Learning for AUC Maximization via Surrogate Loss

## Abstract

In classification tasks, the area under the ROC curve (AUC) is a key metric for evaluating a model's ability to discriminate between positive and negative samples. An AUC-maximizing classifier can have significant advantages in cases where ranking correctness is valued or when the outcome is rare. While ensemble learning is a common strategy to improve predictive performance by combining multiple base models, direct AUC maximization for aggregating base learners leads to an NP-hard optimization challenge. To address this challenge, we propose a novel stacking framework that leverages a linear combination of base models through a surrogate loss function designed to maximize AUC. Our approach learns data-driven stacking weights for base models by minimizing a pairwise loss-based objective. Theoretically, we prove that the resulting ensemble is asymptotically optimal with respect to AUC. Moreover, when the set of base models includes correctly specified models, our method asymptotically concentrates all weight on these models, ensuring consistency. In numerical simulations, the proposed method reduces the AUC risk by up to 20% compared to existing ensemble methods, a finding that is corroborated by real-data analysis, which also shows a reduction of over 30%.

## 1 Introduction

The Receiver Operating Characteristic (ROC) curve is a fundamental tool in machine learning for evaluating the trade-off between sensitivity and specificity in classification models. The area under the ROC curve (AUC) serves as a widely-adopted metric for assessing the generalization performance of classifiers, particularly valued for its robustness to class imbalance. While classification accuracy remains a common evaluation measure, it can be misleading under class distribution skew, whereas AUC provides a more reliable indicator of model discrimination capability for imbalanced data. Moreover, Huang & Ling (2005) theoretically and empirically demonstrated the superiority of AUC over accuracy when selecting metrics for classification model predictive performance.

Over the past two decades, a plethora of research aimed at optimizing AUC has emerged, including representative works on full-batch optimization methods (Yan et al., 2003; Freund et al., 2003; Joachims, 2005), online incremental learning methods (Gao et al., 2013), stochastic optimization methods (Ying et al., 2016; Liu et al., 2018), and more recently, deep learning methods (Liu et al., 2020; Yuan et al., 2021), etc. A comprehensive survey of recent advances in AUC optimization can be found in (Yang & Ying, 2022).

In practical applications, practitioners often have access to multiple base models, raising the important question of how to best leverage these models to achieve optimal predictive performance. Ensemble learning emerges as a natural strategy to address this challenge, with stacking representing a particularly powerful approach that trains a meta-model to optimally combine predictions from diverse base learners. By transforming base model outputs into meta-features through cross-validation, stacking enables the meta-learner to capture complex relationships between these predictions and the target variable.

However, a significant challenge persists: even within the stacking framework which linearly combines base models, direct optimization of AUC for determining model combination weights leads to an NP-hard optimization problem. This computational intractability motivates our proposed ap-

proach of employing a convex and differentiable surrogate loss function to approximate AUC optimization.

In this paper, we propose an **E**nsemble **L**earning method for **AUC M**aximization (**ELAM**), a novel stacking-based framework that maximizes AUC through surrogate loss optimization. Our method determines data-driven stacking weights by minimizing a $K$-fold cross-validation objective based on pairwise surrogate loss, without imposing restrictions on base model structures. Our contributions can be summarized as follows:

- We address the NP-hard challenge of direct AUC optimization for ensemble weighting by proposing a surrogate loss-based approach within a stacking framework.

- We establish theoretical guarantees demonstrating that our method achieves asymptotic optimality in both surrogate risk minimization and AUC maximization.

- We prove that when correctly specified models exist among the base learners, ELAM asymptotically concentrates stacking weight allocation on these optimal models.

- Empirical evaluations on real-world datasets and simulation data demonstrate significant improvements in AUC performance compared to competing methods.

- To the best of our knowledge, this represents the first work to provide theoretical foundations for AUC maximization in ensemble learning methods.

## 2 PRELIMINARIES

### 2.1 STACKING FRAMEWORK AND AUC MAXIMIZATION

Consider $n$ independent and identically distributed (i.i.d.) observations $D_n = \{(y_i, \boldsymbol{x}_i); i = 1, \ldots, n\}$, where the joint distribution $\mathcal{D}$ of $(y_i, \boldsymbol{x}_i)$ is unknown. $\boldsymbol{x}_i \in \mathbb{R}^q$ denotes the feature vector with unknown marginal distribution $\mathcal{P}$. The binary label $y_i \in \{0, 1\}$, where $y_i = 1$ indicates a positive instance and 0 otherwise, with $\Pr(y_i = 1) = p$ and $p$ remains unknown. The likelihood function for these $n$ observations is:

$$L(\boldsymbol{\theta}) = \prod_{i=1}^{n} P(y_i|\boldsymbol{x}_i, \boldsymbol{\theta}),$$

where $P$ is an unknown conditional probability distribution function and $\boldsymbol{\theta}$ is an unknown parameter vector.[1]

We aim to predict new observations while maximizing the area under the ROC curve (AUC) without restricting the structure of base models. Consider $M$ base models, each characterized by a quasi-likelihood function:

$$\prod_{i=1}^{n} P_{(m)}(y_i|\boldsymbol{x}_i, \boldsymbol{\theta}_{(m)}), \tag{1}$$

where $\boldsymbol{\theta}_{(m)}$ denotes the parameter vector for the $m$-th model and it is unknown. We allow $P_{(m)}$ to be mis-specified, meaning $P_{(m)}$ may differ from the true conditional distribution $P$.

Let $\hat{\boldsymbol{\theta}}_{(m)}$ be the parameter estimate obtained by fitting the $m$-th model to $D_n$. Define the weight vector $\boldsymbol{w} = (w_1, \ldots, w_M)^\top$, whose components satisfy $0 \leq w_m \leq C$, i.e., $\boldsymbol{w} \in \mathcal{W} = \{\boldsymbol{w} \in \mathbb{R}^M : \boldsymbol{w} \in [0, C]^M\}$, where $C$ is a constant. For a new observation $\boldsymbol{x}_{n+1}$, the weighted combination of the predicted probabilities from each base model, $P_{(m)}(y_{n+1} = 1|\boldsymbol{x}_{n+1}, \hat{\boldsymbol{\theta}}_{(m)})$, yields the stacking prediction of the new observation:

$$\hat{P}_{\boldsymbol{x}_{n+1}}(\boldsymbol{w}) = \sum_{m=1}^{M} w_m P_{(m)}(y_{n+1} = 1|\boldsymbol{x}_{n+1}, \hat{\boldsymbol{\theta}}_{(m)}) = \sum_{m=1}^{M} w_m \hat{P}_{(m), \boldsymbol{x}_{n+1}} = \boldsymbol{w}^\top \hat{\boldsymbol{P}}_{\boldsymbol{x}_{n+1}}, \tag{2}$$

where $\hat{\boldsymbol{P}}_{\boldsymbol{x}_{n+1}} = (\hat{P}_{(1), \boldsymbol{x}_{n+1}}, \ldots, \hat{P}_{(M), \boldsymbol{x}_{n+1}})^\top$ collects the base model predictions.

---

[1]This paper adopts the convention: $P$ denotes conditional probability, $\Pr$ denotes marginal probability.

For a classifier $f$ that produces real-valued classification scores, we assume that for any two independent samples $(y, \boldsymbol{x})$ and $(y', \boldsymbol{x}')$, $\Pr\big(f(\boldsymbol{x}) = f(\boldsymbol{x}')\big) = 0$. The AUC is defined as:

$$\text{AUC}(f) = \mathbb{E}_{\boldsymbol{x}^+ \sim \mathcal{P}^+, \boldsymbol{x}^- \sim \mathcal{P}^-}[\mathbb{I}\{f(\boldsymbol{x}^+) - f(\boldsymbol{x}^-) \geq 0\}], \tag{3}$$

where $\boldsymbol{x}^+ \sim \mathcal{P}^+$ and $\boldsymbol{x}^- \sim \mathcal{P}^-$ denote that $\boldsymbol{x}^+$ and $\boldsymbol{x}^-$ are independently distributed random samples from the positive and negative classes, respectively.

We use AUC risk to evaluate prediction models. The corresponding AUC risk is $R(f) = 1 - \text{AUC}(f)$, with non-parametric empirical estimator:

$$\hat{R}(f) = \frac{1}{N_+ N_-} \sum_{i=1}^{N_+} \sum_{j=1}^{N_-} \mathbb{I}\{f(\boldsymbol{x}_i^+) - f(\boldsymbol{x}_j^-) < 0\}. \tag{4}$$

In Eq. 4 , we partition all observations $D_n$ into positive and negative sample subsets $\mathcal{S}^+$ and $\mathcal{S}^-$ based on the label $y$: $\mathcal{S}^+ = \{(y_1^+, \boldsymbol{x}_1^+), \ldots, (y_{N_+}^+, \boldsymbol{x}_{N_+}^+)\}$, $\mathcal{S}^- = \{(y_1^-, \boldsymbol{x}_1^-), \ldots, (y_{N_-}^-, \boldsymbol{x}_{N_-}^-)\}$, where $N_+$ and $N_-$ represent the number of positive and negative samples in the $n$ samples, respectively, and $\sum_{i=1}^{N_+} \sum_{j=1}^{N_-}$ denotes summation over all positive-negative sample pairs.

According to the above definitions, maximizing AUC is equivalent to minimizing AUC risk. Therefore, these two will not be distinguished hereafter. Our objective is to determine weights $\boldsymbol{w}$ that maximize the ensemble AUC:

$$\text{AUC}(\boldsymbol{w}) = \mathbb{E}_{\boldsymbol{x}^+ \sim \mathcal{P}^+, \boldsymbol{x}^- \sim \mathcal{P}^-}[\mathbb{I}\{\boldsymbol{w}^\top \hat{\boldsymbol{P}}_{\boldsymbol{x}^+} - \boldsymbol{w}^\top \hat{\boldsymbol{P}}_{\boldsymbol{x}^-} \geq 0\}], \tag{5}$$

where the expectation operator $\mathbb{E}$ covers all random variables in the expression. This notation will be used throughout the remainder of this paper, wherever no confusion arises.

Similarly, maximizing Eq. 5 is equivalent to minimizing:

$$R(\boldsymbol{w}) = \mathbb{E}_{\boldsymbol{x}^+ \sim \mathcal{P}^+, \boldsymbol{x}^- \sim \mathcal{P}^-}[\mathbb{I}\{\boldsymbol{w}^\top \hat{\boldsymbol{P}}_{\boldsymbol{x}^+} - \boldsymbol{w}^\top \hat{\boldsymbol{P}}_{\boldsymbol{x}^-} < 0\}]. \tag{6}$$

Ideally, one could derive the optimal weight by directly maximizing Eq. 5, but this is infeasible as it depends on the unknown distribution of random samples. We therefore propose a surrogate loss approach based on $K$-fold cross-validation. Furthermore, note that both $\text{AUC}(\boldsymbol{w})$ and $R(\boldsymbol{w})$ defined in Eq. 5 and 6 are scale-invariant, satisfying $\text{AUC}(\boldsymbol{w}) = \text{AUC}(\frac{\boldsymbol{w}}{\|\boldsymbol{w}\|_1})$ and $R(\boldsymbol{w}) = R(\frac{\boldsymbol{w}}{\|\boldsymbol{w}\|_1})$ for any $\boldsymbol{w} \in \mathcal{W}$, where $\|\boldsymbol{w}\|_1 = \sum_{m=1}^M |w_m|$.

## 2.2 RELATED WORKS

**Ensemble learning in statistics.** In statistics, model averaging represents a closely related ensemble approach that linearly combines base model predictions. Early work by Bates & Granger (1969) applied model averaging to airline demand forecasting, while Buckland et al. (1997) introduced smoothed AIC and BIC weighting schemes. A significant advancement came from Hansen (2007), who established asymptotic optimality for Mallows model averaging. Subsequent research has extended these ideas to various settings (Zhang et al., 2016; Feng et al., 2024b; Wang et al., 2024; You et al., 2024; Zhang & Liu, 2023; Yu et al., 2025). However, theoretical foundations for AUC-optimal ensemble methods in classification remain underdeveloped.

**Surrogate objectives for AUC maximization.** To circumvent the NP-hard nature of direct AUC optimization, one often replaces the indicator function in the non-parametric estimators of AUC risk defined above by a surrogate loss function $\phi(f(\boldsymbol{x}^+) - f(\boldsymbol{x}^-))$ of $\mathbb{I}\{f(\boldsymbol{x}^+) - f(\boldsymbol{x}^-) \leq 0\}$ to formulate the objective function (Yang & Ying, 2022). As a result, we can define the surrogate objective as

$$\hat{R}(f) = \frac{1}{N_+ N_-} \sum_{i=1}^{N_+} \sum_{j=1}^{N_-} \phi(f(\boldsymbol{x}_i^+) - f(\boldsymbol{x}_j^-)). \tag{7}$$

Gao & Zhou (2015) proposed the definition of AUC consistency for surrogate loss functions: For any distribution $\mathcal{D}$ of samples $(y, \boldsymbol{x})$, for any sequence of classifiers $\{f^{\langle n \rangle}(\boldsymbol{x})\}_{n \geq 1}$:

$$\text{if } R_\phi(f^{\langle n \rangle}) \to \inf_{f \in \sigma(\boldsymbol{x})} R_\phi(f), \text{ then } R(f^{\langle n \rangle}) \to \inf_{f \in \sigma(\boldsymbol{x})} R(f),$$

where $\sigma(\boldsymbol{x})$ represents the set of all measurable functions with respect to $\boldsymbol{x}$. They identified that exponential loss $\phi(x) = e^{-x}$ and logistic loss $\phi(x) = \ln(1 + e^{-x})$ are consistent with AUC. Subsequently, Gao et al. (2013) proved that $\phi(x) = (1 - x)^2$ is also consistent.

**Research on AUC in Ensemble Learning.** Existing ensemble methods for AUC maximization are as follows. LeDell et al. (2016) proposed a stacking method that, based on $K$-fold cross-validation, derives the stacking weights of base models by minimizing the average empirical AUC risk of the ensemble model on K validation sets. Figini et al. (2016) proposed a stacking method that weights base models based on their non-parametric estimator of AUC values. However, these methods lack theoretical guarantees, and the former method will lead to an NP-hard optimization problem.

## 3    STACKING METHOD BASED ON $K$-FOLD CROSS-VALIDATION

We now present our proposed method, **ELAM** (Ensemble Learning for AUC Maximization), which leverages $K$-fold cross-validation to determine optimal combination weights by minimizing a pairwise surrogate loss objective.

### 3.1    CROSS-VALIDATION SCHEME

The $K$-fold cross-validation procedure for generating out-of-sample predictions proceeds as follows:

- Randomly partition the dataset into $K$ mutually exclusive folds of equal size, where $2 \leq K \leq n$, and each fold contains $J = n/K$ observations.
- For $k = 1, \ldots, K$,
    - Designate the $k$-th fold as the validation set, with the remaining $K - 1$ folds forming the training set $D_n^{[-k]}$.
    - Train each base model $m = 1, \ldots, M$ on $D_n^{[-k]}$ to obtain parameter estimates $\hat{\boldsymbol{\theta}}_{(m)}^{[-k]}$.
    - Generate predictions for the validation samples using the trained models. For the $j$-th observation in the $k$-th fold ($j = 1, \ldots, J$), the prediction from base model $m$ is:
    $$\tilde{P}_{(m),j}^{[-k]} = P_{(m)}\left(y_{(k-1)\times J+j} = 1 \mid \boldsymbol{x}_{(k-1)\times J+j}, \hat{\boldsymbol{\theta}}_{(m)}^{[-k]}\right).$$

    The combined vector of prediction from $M$ base models for this observation is denoted as:
    $$\tilde{\boldsymbol{P}}_{\boldsymbol{x}_{(k-1)\times J+j}}^{[-k]} = (\tilde{P}_{(1),j}^{[-k]}, \ldots, \tilde{P}_{(M),j}^{[-k]})^{\top}. \tag{8}$$
- Aggregate the cross-validation predictions across all folds to form the complete set of out-of-sample predictions for each base model.

This procedure ensures that each observation is predicted by models trained on independent data, providing unbiased estimates of base model performance.

### 3.2    SURROGATE LOSS OPTIMIZATION

The ideal objective for AUC maximization would minimize the empirical AUC risk:

$$CV_K(\boldsymbol{w}) = \frac{1}{N_+N_-}\sum_{i=1}^{N_+}\sum_{j=1}^{N_-}\mathbb{I}\{\boldsymbol{w}^{\top}\tilde{\boldsymbol{P}}_{\boldsymbol{x}_i^+} - \boldsymbol{w}^{\top}\tilde{\boldsymbol{P}}_{\boldsymbol{x}_j^-} < 0\}, \tag{9}$$

where $\tilde{\boldsymbol{P}}_{\boldsymbol{x}_i^+}$ and $\tilde{\boldsymbol{P}}_{\boldsymbol{x}_j^-}$ denote the cross-validation predictions for positive and negative instances defined in Eq. 8, respectively. The strict notation for $\tilde{\boldsymbol{P}}_{\boldsymbol{x}_i^+}$ and $\tilde{\boldsymbol{P}}_{\boldsymbol{x}_j^-}$ should be $\tilde{\boldsymbol{P}}_{\boldsymbol{x}_i^+}^{[-\sigma(i)]}$ and $\tilde{\boldsymbol{P}}_{\boldsymbol{x}_j^-}^{[-\tau(j)]}$, where $\sigma(i)$ is a mapping indicating that $\boldsymbol{x}_i^+$ belongs to the $\sigma(i)$-th fold in cross-validation, and $\tau(j)$ similarly indicates that $\boldsymbol{x}_j^-$ belongs to the $\tau(j)$-th fold. Henceforth we simply write $\tilde{\boldsymbol{P}}_{\boldsymbol{x}_i^+}$ and $\tilde{\boldsymbol{P}}_{\boldsymbol{x}_j^-}$ hereafter, wherever no confusion arises.

However, this objective involves the non-convex indicator function, leading to an NP-hard optimization problem. To overcome this problem, we employ a smooth, differentiable and convex function

$\phi : \mathbb{R} \to \mathbb{R}^+$ that approximates the indicator function. The resulting surrogate risk for the stacking classifier is:

$$R_\phi(\boldsymbol{w}) = \mathbb{E}_{\boldsymbol{x}^+ \sim \mathcal{P}^+, \boldsymbol{x}^- \sim \mathcal{P}^-} \left[ \phi \left( \boldsymbol{w}^\top \hat{\boldsymbol{P}}_{\boldsymbol{x}^+} - \boldsymbol{w}^\top \hat{\boldsymbol{P}}_{\boldsymbol{x}^-} \right) \right]. \tag{10}$$

Accordingly, we define the cross-validation objective using the logistic loss $\phi(x) = \ln(1 + e^{-x})$, which enjoys established consistency properties for AUC optimization (Gao & Zhou, 2015):

$$CV_\phi^K(\boldsymbol{w}) = \frac{1}{N_+ N_-} \sum_{i=1}^{N_+} \sum_{j=1}^{N_-} \phi \left( \boldsymbol{w}^\top \tilde{\boldsymbol{P}}_{\boldsymbol{x}_i^+} - \boldsymbol{w}^\top \tilde{\boldsymbol{P}}_{\boldsymbol{x}_j^-} \right), \tag{11}$$

while using $\phi(x) = e^{-x}$ or $\phi(x) = (1-x)^2$ would yield similar results.

The optimal stacking weights are obtained by solving the constrained optimization problem:

$$\hat{\boldsymbol{w}} = \arg\min_{\boldsymbol{w} \in \mathcal{W}} CV_\phi^K(\boldsymbol{w}),$$

where $\mathcal{W} = \{\boldsymbol{w} \in \mathbb{R}^M : \boldsymbol{w} \in [0, C]^M\}$. For a new observation $\boldsymbol{x}_{n+1}$, the final prediction combines base model outputs using the optimized stacking weights:

$$\hat{P}_{\boldsymbol{x}_{n+1}}(\hat{\boldsymbol{w}}) = \sum_{m=1}^{M} \hat{w}_m P_{(m)}\big(y_{n+1} = 1 \big| \boldsymbol{x}_{n+1}, \hat{\boldsymbol{\theta}}_{(m)}\big). \tag{12}$$

We summarize our method in Algorithm 1.

---

**Algorithm 1 ELAM**: Ensemble Learning for AUC Maximization

| | |
|---|---|
| **Input** | Dataset $D_n = \{(y_i, \boldsymbol{x}_i)\}_{i=1}^n$; base models $\{P_{(m)}\}_{m=1}^M$; number of folds $K$; surrogate loss $\phi$, new observation $\boldsymbol{x}_{n+1}$. |
| **Cross-Validate** | **for** $k = 1$ to $K$ **do** |
| | $\quad$ Train each $P_{(m)}$ on $D_n^{[-k]}$ to get $\hat{\boldsymbol{\theta}}_{(m)}^{[-k]}$ |
| | $\quad$ Predict on fold $k$: $\tilde{P}_{(m),j}^{[-k]} = P_{(m)}(y_j = 1 | \boldsymbol{x}_j, \hat{\boldsymbol{\theta}}_{(m)}^{[-k]})$ |
| | **end for** |
| | Aggregate $\tilde{\boldsymbol{P}}_{\boldsymbol{x}_i}$ for all $i = 1, \ldots, n$ |
| **Optimize** | Compute $CV_\phi^K(\boldsymbol{w}) = \frac{1}{N_+ N_-} \sum_{i=1}^{N_+} \sum_{j=1}^{N_-} \phi \left( \boldsymbol{w}^\top \tilde{\boldsymbol{P}}_{\boldsymbol{x}_i^+} - \boldsymbol{w}^\top \tilde{\boldsymbol{P}}_{\boldsymbol{x}_j^-} \right)$ |
| | Solve $\hat{\boldsymbol{w}} = \arg\min_{\boldsymbol{w} \in \mathcal{W}} CV_\phi^K(\boldsymbol{w})$ |
| **Finally Train** | **for** $m = 1$ to $M$ **do** |
| | $\quad$ Train $P_{(m)}$ on full $D_n$ to get $\hat{\boldsymbol{\theta}}_{(m)}$ |
| | **end for** |
| **Predict** | Obtain the prediction $\hat{P}_{\boldsymbol{x}_{n+1}}(\hat{\boldsymbol{w}}) = \sum_{m=1}^{M} \hat{w}_m P_{(m)}\big(y_{n+1} = 1 \big| \boldsymbol{x}_{n+1}, \hat{\boldsymbol{\theta}}_{(m)}\big)$ |
| **Output** | Return $\hat{P}_{\boldsymbol{x}_{n+1}}(\hat{\boldsymbol{w}})$ |

---

In finite samples, prediction results can be sensitive to the value of $K$, especially when $K$ is small. When both $K$ and $n$ are large, computational costs increase significantly. Following common practice in cross-validation literature, we set $K = 10$ throughout our experiments, balancing computational efficiency with estimation accuracy. The selection of $K$ is detailed in Zhang & Liu (2023) and will not be discussed further here.

## 4 THEORETICAL ANALYSIS

This section establishes the asymptotic properties of the ELAM method. We analyze both the surrogate risk $R_\phi(\boldsymbol{w})$ and the AUC risk $R(\boldsymbol{w})$ of the stacking predictor, providing theoretical guarantees for its optimality and consistency. We begin by stating the assumptions required for our theoretical analysis. The limiting process referred to is in the sense of $n \to \infty$.

## 4.1 Asymptotic Optimality under Surrogate Risk

**Assumption 1.** *Let $M \leq n$. The limit value of $\hat{\boldsymbol{\theta}}_{(m)}$ is $\boldsymbol{\theta}^*_{(m)}$, i.e., for $\forall m \in \{1, \ldots, M\}$, $\hat{\boldsymbol{\theta}}_{(m)} - \boldsymbol{\theta}^*_{(m)} = O_p(n^{-1/2}M^{1/2})$ holds. For $\hat{\boldsymbol{\theta}}^{[-k]}_{(m)}$, since $n$ and $n - n/K$ are of the same order, we also have $\hat{\boldsymbol{\theta}}^{[-k]}_{(m)} - \boldsymbol{\theta}^*_{(m)} = O_p(n^{-1/2}M^{1/2})$.*

**Remark:** Assumption 1 requires that the estimator of each base model's parameter $\boldsymbol{\theta}_{(m)}$ converges to some limit value $\boldsymbol{\theta}^*_{(m)}$ at a certain rate. This condition is commonly used when studying the asymptotic properties of nonlinear model averaging estimators in statistics. Zhang et al. (2016) assumed a rate of $O_p(n^{-1/2})$; Zhang & Liu (2023) allowed the number of base models $M$ to diverge and assumed a rate of $O_p(n^{-1/2}M^{1/2})$.

**Assumption 2.** *For $m = 1, \ldots, M, k = 1, \ldots, K, j = 1, \ldots, J$, $\tilde{P}^{[-k]}_{(m),j}$ is differentiable with respect to $\hat{\boldsymbol{\theta}}^{[-k]}_{(m)}$, and there exists a constant $\varrho > 0$ such that the following holds uniformly for $m = 1, \ldots, M$:*

$$\mathbb{E} \sup_{\boldsymbol{\theta}^\star \in \mathcal{O}(\boldsymbol{\theta}^*_{(m)}, \varrho)} \left\| \frac{\partial \tilde{P}^{[-k]}_{(m),j}}{\partial \hat{\boldsymbol{\theta}}^{[-k]}_{(m)}} \Big|_{\hat{\boldsymbol{\theta}}^{[-k]}_{(m)} = \boldsymbol{\theta}^\star} \right\|^2 = O(1),$$

*where $\mathcal{O}(\boldsymbol{\theta}^*_{(m)}, \varrho)$ denotes a neighborhood centered at $\boldsymbol{\theta}^*_{(m)}$ with radius $\varrho$.*

**Assumption 3.** $n/N_+ = O(1)$, $n/N_- = O(1)$.

**Remark:** Assumption 2 requires that the base model estimators are differentiable and their gradients are bounded, which is also assumed in Zhang & Liu (2023) and Feng et al. (2024a). Assumption 3 requires that the ratio of positive to negative samples among the $n$ observations be bounded away from zero, equivalently, there exists a constant $c \in (0, 1)$ such that $\frac{\min\{N_+, N_-\}}{n} \geq c$.

For a new observation $\boldsymbol{x}_{n+1}$, the prediction of the $m$-th base model at the limit value of parameter $\boldsymbol{\theta}^*_{(m)}$ is $P^*_{(m), \boldsymbol{x}_{n+1}} = P_{(m)}(y_{n+1} = 1 | \boldsymbol{x}_{n+1}, \boldsymbol{\theta}^*_{(m)})$. The stacking prediction for the new observation $\boldsymbol{x}_{n+1}$ at the limit values is:

$$P^*_{\boldsymbol{x}_{n+1}}(\boldsymbol{w}) = \sum_{m=1}^{M} w_m P^*_{(m), \boldsymbol{x}_{n+1}}. \tag{13}$$

Denote the surrogate risk at the limit values as: $R^*_\phi(\boldsymbol{w}) = \mathbb{E}_{\boldsymbol{x}^+ \sim \mathcal{P}^+, \boldsymbol{x}^- \sim \mathcal{P}^-}[\phi(P^*_{\boldsymbol{x}^+}(\boldsymbol{w}) - P^*_{\boldsymbol{x}^-}(\boldsymbol{w}))]$, where $P^*_{\boldsymbol{x}^+}(\boldsymbol{w})$ and $P^*_{\boldsymbol{x}^-}(\boldsymbol{w})$ are defined as in Eq. 13. Let $\xi_n = \inf_{\boldsymbol{w} \in \mathcal{W}} R^*_\phi(\boldsymbol{w})$ be the infimum of the surrogate risk of the stacking predictor at the limit values.

**Assumption 4.** $Cn^{-1/2}M^{3/2}\xi_n^{-1} = o(1)$.

**Remark:** Assumption 4 puts a bound on the number of models relative to the sample size, and it specifies that $M$ grows at a rate no faster than $C^{-2/3}n^{1/3}\xi_n^{2/3}$. Compared to the existing literature on model averaging in statistics, assumption 4 is stricter than the common conditions $n^{-1/2}\xi_n^{-1} = o(1)$(Zhang et al., 2016)[2] or $n^{-1/2}M\xi_n^{-1} = o(1)$(Zhang & Liu, 2023). This is because in this work, we extend the weight space from $\{\boldsymbol{w} \in [0,1]^M : \sum_{m=1}^{M} w_m = 1\}$ to $\{\boldsymbol{w} \in \mathbb{R}^M : \boldsymbol{w} \in [0, C]^M\}$, which necessitates stricter conditions to establish the asymptotic optimality of the stacking prediction.

**Assumption 5.** $\xi_n^{-1} \sup_{\boldsymbol{w} \in \mathcal{W}} \left[ \phi(\hat{P}_{\boldsymbol{x}^+}(\boldsymbol{w}) - \hat{P}_{\boldsymbol{x}^-}(\boldsymbol{w})) - \phi(P^*_{\boldsymbol{x}^+}(\boldsymbol{w}) - P^*_{\boldsymbol{x}^-}(\boldsymbol{w})) \right]$ *is uniformly integrable.*

**Remark:** Assumption 5 is not an intuitive condition. In proving Theorem 1 we show that $\xi_n^{-1} \sup_{\boldsymbol{w} \in \mathcal{W}} \left[ \phi(\hat{P}_{\boldsymbol{x}^+}(\boldsymbol{w}) - \hat{P}_{\boldsymbol{x}^-}(\boldsymbol{w})) - \phi(P^*_{\boldsymbol{x}^+}(\boldsymbol{w}) - P^*_{\boldsymbol{x}^-}(\boldsymbol{w})) \right] = o_p(1)$. This assumption guarantees that the expectation of the left-hand side is $o(1)$.

---

[2]The $\xi_n$ defined in Zhang et al. (2016) is based on the loss of $n$ sample points. The original condition was $n^{1/2}\xi_n^{-1} = o(1)$; the condition listed here, $n^{-1/2}\xi_n^{-1} = o(1)$, is the result after eliminating the effect of sample size.

**Assumption 6.** *Let $t(\boldsymbol{w}) = CV_\phi^*(\boldsymbol{w})/R_\phi^*(\boldsymbol{w}) - 1$. There exists $\kappa_T = O_p(1)$ such that for $\forall \boldsymbol{w}, \boldsymbol{w}' \in \mathcal{W}$, $|t(\boldsymbol{w}) - t(\boldsymbol{w}')| \leq \kappa_T \|\boldsymbol{w} - \boldsymbol{w}'\|_1$ holds.*

**Remark:** Assumption 6 ensures the stochastic equicontinuity of $t(\boldsymbol{w})$ with respect to $\boldsymbol{w}$ (Newey, 1991). Yu et al. (2025) first used the concept of stochastic equicontinuity to prove the asymptotic optimality of model averaging methods in statistics. This assumption is similar to Condition 4 in Gao et al. (2023) and Assumption 3 in Yu et al. (2025).

**Theorem 1.** *Under Assumptions 1 - 5, the stacking weight $\hat{\boldsymbol{w}}$ derived by the ELAM method satisfies:*

$$\frac{R_\phi(\hat{\boldsymbol{w}})}{\inf_{\boldsymbol{w} \in \mathcal{W}} R_\phi(\boldsymbol{w})} \xrightarrow{p} 1.$$

**Remark:** Theorem 1 establishes that ELAM achieves asymptotic optimality with respect to the surrogate-risk objective; that is, the prediction built with the optimal stacking weight $\hat{\boldsymbol{w}}$ asymptotically attains the theoretical infimum of the surrogate risk over the stacking-predictor class.

4.2 ASYMPTOTIC OPTIMALITY UNDER AUC RISK

Theorem 1 establishes the asymptotic optimality of the ELAM method in terms of surrogate risk. Below, we further establish its asymptotic optimality in terms of AUC risk. Denote the AUC risk at the limit values as $R^*(\boldsymbol{w}) = \mathbb{E}_{\boldsymbol{x}^+ \sim \mathcal{P}^+, \boldsymbol{x}^- \sim \mathcal{P}^-}[\mathbb{I}\{P_{\boldsymbol{x}^+}^*(\boldsymbol{w}) - P_{\boldsymbol{x}^-}^*(\boldsymbol{w}) < 0\}]$, and let $\xi_n^* = \inf_{\boldsymbol{w} \in \mathcal{W}} R^*(\boldsymbol{w})$ be the infimum of the AUC risk of the stacking predictor at the limit values.

**Assumption 7.** *For any new sample points $(y, \boldsymbol{x})$, $(y', \boldsymbol{x}')$, $f_{a(\boldsymbol{w})|\Delta(\boldsymbol{w})}(x) = \frac{\partial F_{a(\boldsymbol{w})|\Delta(\boldsymbol{w})}(x)}{\partial x}$ is uniformly bounded, $F_{a(\boldsymbol{w})|\Delta(\boldsymbol{w})}(x)$ is the cumulative distribution function of $a(\boldsymbol{w})$ given $\Delta(\boldsymbol{w})$, and $\xi_n^{-1} \sup_{\boldsymbol{w} \in \mathcal{W}}\{P(a(\boldsymbol{w}) + \Delta(\boldsymbol{w}) < 0 \mid \Delta(\boldsymbol{w})) - P(a(\boldsymbol{w}) < 0 \mid \Delta(\boldsymbol{w}))\}$ is uniformly integrable, where $a(\boldsymbol{w}) = (y - y') \sum_{m=1}^M w_m(P_{(m),\boldsymbol{x}}^* - P_{(m),\boldsymbol{x}'}^*)$ and*

$$\Delta(\boldsymbol{w}) = (y-y') \sum_{m=1}^M w_m \left[ \left(\hat{\boldsymbol{\theta}}_{(m)} - \boldsymbol{\theta}_{(m)}^*\right)^\top \frac{\partial \hat{P}_{(m),\boldsymbol{x}}}{\partial \hat{\boldsymbol{\theta}}_{(m)}}|_{\hat{\boldsymbol{\theta}}_{(m)} = \boldsymbol{\theta}_{(m),\boldsymbol{x}}^\star} - \left(\hat{\boldsymbol{\theta}}_{(m)} - \boldsymbol{\theta}_{(m)}^*\right)^\top \frac{\partial \hat{P}_{(m),\boldsymbol{x}'}}{\partial \hat{\boldsymbol{\theta}}_{(m)}}|_{\hat{\boldsymbol{\theta}}_{(m)} = \boldsymbol{\theta}_{(m),\boldsymbol{x}'}^\star} \right],$$

*where $\boldsymbol{\theta}_{(m),\boldsymbol{x}}^\star, \boldsymbol{\theta}_{(m),\boldsymbol{x}'}^\star \in \mathcal{O}(\boldsymbol{\theta}_{(m)}^*, \varrho)$.*

**Remark:** Assumption 7 is similar to Assumption 6 in Feng et al. (2024a). This assumption ensures that we can derive: $\xi_n^{-1} \sup_{\boldsymbol{w} \in \mathcal{W}}\{P(a(\boldsymbol{w}) + \Delta(\boldsymbol{w}) < 0 \mid \Delta(\boldsymbol{w})) - P(a(\boldsymbol{w}) < 0 \mid \Delta(\boldsymbol{w}))\} = o_p(1)$, and that the result after taking the expectation of this expression is $o(1)$.

**Assumption 8.** *For $\forall \boldsymbol{w} \in \mathcal{W}$, $R^*(\boldsymbol{w}) - \inf_{\boldsymbol{w} \in \mathcal{W}} R^*(\boldsymbol{w}) \leq 2\sqrt{R_\phi^*(\boldsymbol{w}) - \inf_{\boldsymbol{w} \in \mathcal{W}} R_\phi^*(\boldsymbol{w})}$, and if $R_\phi^*(\hat{\boldsymbol{w}}) - \inf_{\boldsymbol{w} \in \mathcal{W}} R_\phi^*(\boldsymbol{w}) = o_p(1)$, then $2\xi_n^{*-1}\sqrt{R_\phi^*(\hat{\boldsymbol{w}}) - \inf_{\boldsymbol{w} \in \mathcal{W}} R_\phi^*(\boldsymbol{w})} = o_p(1)$.*

**Assumption 9.** $Cn^{-1/2}M^{3/2}\xi_n^{*-1} = o(1)$.

**Remark:** Gao & Zhou (2015) proved that for the logistic loss $\phi(x) = \ln(1 + e^{-x})$, the surrogate risk and the AUC risk of any classifier $f$ satisfy: $R^*(f) - \inf_{f \in \sigma(\boldsymbol{x})} R^*(f) \leq 2\sqrt{R_\phi^*(f) - \inf_{f \in \sigma(\boldsymbol{x})} R_\phi^*(f)}$. The first part of Assumption 8 guarantees that this inequality remains valid when the function space is restricted to the stacking class. The second part ensures that the rate at which $\xi_n^*$ converges to 0 is bounded above by $\sqrt{R_\phi^*(\hat{\boldsymbol{w}}) - \inf_{\boldsymbol{w} \in \mathcal{W}} R_\phi^*(\boldsymbol{w})}$, a condition similar to Assumption 7 in Feng et al. (2024a). Assumption 9 is similar to Assumption 4 and will not be elaborated further. The upper-bound restriction $C$ on the stacking weight can be made less binding by picking $C$ large, which can be done as long as assumption 4 and 9 hold.

**Theorem 2.** *Under Assumptions 1 - 9, the stacking weight $\hat{\boldsymbol{w}}$ derived by the ELAM method satisfies:*

$$\frac{R(\hat{\boldsymbol{w}})}{\inf_{\boldsymbol{w} \in \mathcal{W}} R(\boldsymbol{w})} \xrightarrow{p} 1.$$

**Remark:** Theorem 2 demonstrates that the asymptotic optimality extends to the original AUC risk objective, providing the primary theoretical guarantee for our method.

## 4.3 Weight Consistency

We now establish consistency properties when correctly specified models exist among the base learners. Following Gao & Zhou (2015), define the optimal classifier class $\mathcal{B} = \{f : (f(\boldsymbol{x}) - f(\boldsymbol{x}')) \times (\eta(\boldsymbol{x}) - \eta(\boldsymbol{x}')) > 0 \text{ if } \eta(\boldsymbol{x}) \neq \eta(\boldsymbol{x}')\}$, where $\eta(\boldsymbol{x}) = P(y = 1 \mid \boldsymbol{x})$. The m-th base model is correctly specified if $P^*_{(m)} \in \mathcal{B}$. Furthermore, let $\boldsymbol{w}^* = \arg\min_{\boldsymbol{w} \in \mathcal{W}} R^*_\phi(\boldsymbol{w})$, i.e., $\boldsymbol{w}^*$ is the weight that minimizes the surrogate risk at the limit values.

**Theorem 3.** *Under Assumptions 1 - 3, 6 and 8, if $w^*$ is unique, and $n^{-1/2} M^{3/2} = o(1)$, then $\boldsymbol{w}^* = \arg\min_{\boldsymbol{w} \in \mathcal{W}} R^*(\boldsymbol{w})$, and the weight $\hat{w}$ derived by the ELAM method satisfies:*

$$\hat{\boldsymbol{w}} \xrightarrow{p} \boldsymbol{w}^*.$$

**Remark:** Theorem 3 establishes that ELAM consistently identifies the optimal weighting scheme. When correctly specified models exist, the normalized weights $\tilde{\boldsymbol{w}} = \hat{\boldsymbol{w}}/\|\hat{\boldsymbol{w}}\|_1$ concentrate on these models, i.e., $\sum_{m \in D} \tilde{w}_m \xrightarrow{p} 1$, where $D$ contains the indices of correctly specified models. This weight consistency property ensures that ELAM asymptotically identifies the best possible ensemble composition.

Furthermore, since the AUC risk of the stacking predictor is scale-invariant, we may use the normalized weight $\tilde{\boldsymbol{w}}$ instead of the original weight $\hat{\boldsymbol{w}}$ to make a prediction for a new instance $\boldsymbol{x}_{n+1}$:

$$\hat{P}_{\boldsymbol{x}_{n+1}}(\tilde{\boldsymbol{w}}) = \sum_{m=1}^{M} \tilde{w}_m P_{(m)}\big(y_{n+1} = 1 \big| \boldsymbol{x}_{n+1}, \hat{\boldsymbol{\theta}}_{(m)}\big). \tag{14}$$

In this case, the stacking prediction $\hat{P}_{\boldsymbol{x}_{n+1}}(\tilde{\boldsymbol{w}})$ is a standard probabilistic prediction.

## 5 Experimental Results

In this section, we evaluate the performance of ELAM on two publicly available benchmark datasets: the Mammographic Mass Dataset Elter (2007) and the Spambase Dataset Hopkins & Suermondt (1999) from the UCI Machine Learning Repository. Furthermore, extra experiments and simulations are provided in the appendix.

### 5.1 Experimental Setup

The Mammographic Mass Dataset comprises 830 observations (after removing missing values from the original 961 instances), each with five clinical features describing mammographic mass lesions. The Spambase Dataset contains 4,601 email samples labeled as spam or non-spam, each characterized by 57 attributes.

For the Mammographic Mass Dataset, we consider the complete non-nested model space consisting of $2^5 - 1 = 31$ distinct feature combinations. For the Spambase Dataset, to maintain computational tractability with the larger feature set, we restrict ourselves to a nested sequence of 57 models obtained by sequentially adding features in their original order. In both cases, we employ logistic regression as the base learner, with model diversity achieved through feature subset selection.

Besides the ELAM method, this paper also considers the following competing methods: (1) Logistic regression model using all covariates, abbreviated as **FULL**; (2) AIC information criterion model selection method, abbreviated as **AIC**; (3) BIC information criterion model selection method, abbreviated as **BIC**; (4) Smoothed AIC information criterion weighted averaging method, abbreviated as **SAIC** (Buckland et al., 1997); (5) Smoothed BIC information criterion weighted averaging method, abbreviated as **SBIC** (Buckland et al., 1997); (6) Model ensemble method proposed by LeDell et al. (2016), in which 10-fold cross-validation is also employed, abbreviated as **ME** (Model ensemble); (7) Stacking method proposed by Figini et al. (2016), abbreviated as **AUCW** (AUC weighted); (8) Simple averaging method that sets the stacking weights of each base model equal, abbreviated as **SA** (Simple averaging). We randomly split the sample into training and testing sets in a 7:3 ratio, and calculate the relative empirical AUC risk of each method through 200 repeated experiments.

## 5.2 RESULTS AND ANALYSIS

Figure 1 presents the empirical AUC risk distributions across all methods on both datasets. ELAM demonstrates consistent superiority, achieving the lowest AUC risk in both experimental settings.

On the Mammographic Mass Dataset, ELAM reduces average AUC risk by substantial margins compared to all competitors: **13.40%** versus ME, **7.87%** versus AUCW, **7.99%** versus SA, **11.62%** versus FULL, **10.98%** versus SAIC, **11.36%** versus SBIC, **10.56%** versus AIC, **11.16%** versus BIC.

The performance advantage remains pronounced on the Spambase Dataset, with relative risk reductions of **23.23%** (ME), **38.64%** (AUCW), **42.22%** (SA), **14.94%** (FULL), **4.45%** (SAIC), **4.47%** (SBIC), **4.50%** (AIC), **4.61%** (BIC).

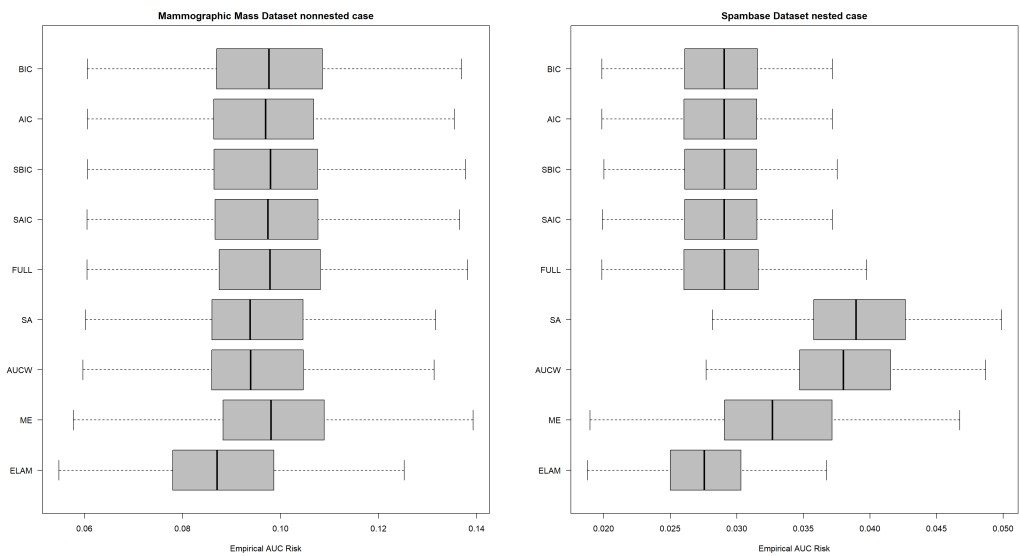

Figure 1: Empirical AUC risk of models on the Mammographic Mass Dataset and Spambase Dataset.

These results demonstrate ELAM's robust performance across different data characteristics and model configurations. The method particularly excels in the non-nested setting (Mammographic Mass), where its ability to intelligently combine diverse base models yields significant gains over selection, averaging and existing ensemble approaches.

## 6 CONCLUSION

This paper has introduced ELAM, a novel stacking framework for AUC maximization in binary classification tasks. Our method addresses the fundamental computational challenge of direct AUC optimization through a surrogate loss approach, while providing strong theoretical guarantees for asymptotic optimality and weight consistency. Real data and numerical simulation results show that the proposed method has significant advantages over other competing methods in maximizing AUC.

While this work provides a solid theoretical and empirical foundation for AUC-optimal ensemble learning, several important directions warrant further investigation. Firstly, the current formulation focuses on binary classification. Extending the framework to multi-class AUC optimization represents an important direction for future research. Secondly, our theoretical analysis assumes fixed parameter dimensions (Assumption 1). Developing extensions that accommodate high-dimensional settings where parameter dimensions grow with sample size would significantly enhance the method's applicability.

Despite these limitations, ELAM represents a significant step forward in ensemble learning methodology, providing both theoretical guarantees and practical benefits for AUC maximization in classification tasks. The framework opens several promising avenues for future research at the intersection of ensemble methods and performance metric optimization.

## 7 ETHICS AND REPRODUCIBILITY STATEMENT

This work presents a methodological contribution in the area of ensemble learning and AUC optimization. The research was conducted in accordance with the ICLR Code of Ethics.

- **Datasets:** Our empirical evaluation utilizes publicly available benchmark datasets from the UCI Machine Learning Repository. These datasets are widely used in the machine learning community for non-commercial research purposes and are pre-anonymized.
- **Compliance:** The proposed method does not present any foreseeable direct negative societal impact. It is designed to improve the ranking performance of classifiers, which is particularly beneficial in domains like medical diagnosis, where accurate ranking of positive instances is crucial.
- **Competing Interests:** The authors declare no competing interests, financial or non-financial, related to this work.
- **Reproducibility:** To ensure the reproducibility, we have provided detailed descriptions of our algorithm, theoretical assumptions, and experimental setup. The use of standard datasets and base learners (logistic regression) further facilitates replication of our results. The code will be released on Github after the double-blind review.

## 8 LLM USAGE STATEMENT

The authors used a large language model (LLM) solely for the purpose of improving the readability and language of this manuscript. Specifically, the LLM was employed to assist with grammar checking, rephrasing for clarity, and ensuring fluency in English. All ideation, theoretical development, algorithmic design, experimental execution, data analysis, and conclusions remain the original work of the authors. The LLM was not used to generate any scientific content, creative ideas, or data interpretations. The authors take full responsibility for the entire content of this paper.

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

# A APPENDIX

## A.1 LEMMA

**Lemma 1** ((Zhang, 2010),(Gao et al., 2019)). *Assume $\mathcal{W}$ is a weight set. If: $\hat{\boldsymbol{w}} = \underset{\boldsymbol{w} \in \mathcal{W}}{\arg\min} \{R(\boldsymbol{w}) + a_n(\boldsymbol{w}) + b_n\}$, where $a_n(\boldsymbol{w})$ is a term related to $\boldsymbol{w}$, and $b_n$ is a term unrelated to $\boldsymbol{w}$. If:*

$$\sup_{\boldsymbol{w} \in \mathcal{W}} |a_n(\boldsymbol{w})|/R^*(\boldsymbol{w}) = o_p(1),$$

$$\sup_{\boldsymbol{w} \in \mathcal{W}} |R^*(\boldsymbol{w}) - R(\boldsymbol{w})|/R^*(\boldsymbol{w}) = o_p(1),$$

*and there exists a constant $c$ and a positive integer $N^*$ such that for $n \geq N^*$, $\inf_{\boldsymbol{w} \in \mathcal{W}} R^*(\boldsymbol{w}) \geq c > 0$ holds almost everywhere, then: $\frac{R(\hat{\boldsymbol{w}})}{\inf_{\boldsymbol{w} \in \mathcal{W}} R(\boldsymbol{w})} \xrightarrow{p} 1$.*

**Lemma 2** ((Hoeffding, 1963)). *Assume $X_1, X_2, \ldots, X_m$ are i.i.d. random variables, $Y_1, Y_2, \ldots, Y_n$ are also i.i.d. random variables, and $X_1, X_2, \ldots, X_m$ and $Y_1, Y_2, \ldots, Y_n$ are mutually independent. Consider a random variable of the form:*

$$U = \frac{1}{mn} \sum_{i=1}^{m} \sum_{j=1}^{n} g(X_i, Y_j),$$

*If $a \leq g \leq b$, then the following holds:*

$$Pr\{|U - EU| \geq t\} \leq 2e^{-2vt^2/(b-a)^2},$$

*where $v = min(m, n)$.*

## A.2 PROOF OF THEOREM 1

In the proofs of this paper, for notational convenience and where no confusion arises, we omit the $K$ in $CV_\phi^K(\boldsymbol{w})$ and denote it simply as $CV_\phi(\boldsymbol{w})$. Note that:

$$\hat{\boldsymbol{w}} = \underset{\boldsymbol{w} \in \mathcal{W}}{\arg\min}\, CV_\phi(\boldsymbol{w}) = \underset{\boldsymbol{w} \in \mathcal{W}}{\arg\min}\, R_\phi(\boldsymbol{w}) + CV_\phi(\boldsymbol{w}) - R_\phi(\boldsymbol{w}),$$

According to Lemma 1, we only need to prove:

$$\sup_{\boldsymbol{w} \in \mathcal{W}} \frac{\left|R_\phi(\boldsymbol{w}) - R_\phi^*(\boldsymbol{w})\right|}{R_\phi^*(\boldsymbol{w})} = o(1), \tag{A.1}$$

$$\sup_{\boldsymbol{w} \in \mathcal{W}} \frac{|CV_\phi(\boldsymbol{w}) - R_\phi(\boldsymbol{w})|}{R_\phi^*(\boldsymbol{w})} = o_p(1), \tag{A.2}$$

If Eq. A.1 is proven, and we also have:

$$\sup_{\boldsymbol{w} \in \mathcal{W}} \frac{\left|CV_\phi(\boldsymbol{w}) - R_\phi^*(\boldsymbol{w})\right|}{R_\phi^*(\boldsymbol{w})} = o_p(1), \tag{A.3}$$

then Eq. A.2 can be directly obtained. Thus, we only need to prove Eq. A.1 and Eq. A.3 .

We first prove Eq. A.1 :

$$\xi_n^{-1} \sup_{\boldsymbol{w} \in \mathcal{W}} |\phi(\hat{P}_{\boldsymbol{x}^+}(\boldsymbol{w}) - \hat{P}_{\boldsymbol{x}^-}(\boldsymbol{w})) - \phi(P_{\boldsymbol{x}^+}^*(\boldsymbol{w}) - P_{\boldsymbol{x}^-}^*(\boldsymbol{w}))|$$

$$\leq \xi_n^{-1} \sup_{\boldsymbol{w} \in \mathcal{W}} \left| \left( \hat{P}_{\boldsymbol{x}^+}(\boldsymbol{w}) - \hat{P}_{\boldsymbol{x}^-}(\boldsymbol{w}) \right) - \left( P_{\boldsymbol{x}^+}^*(\boldsymbol{w}) - P_{\boldsymbol{x}^-}^*(\boldsymbol{w}) \right) \right|$$

$$= \xi_n^{-1} \sup_{\boldsymbol{w} \in \mathcal{W}} \left| \left( \boldsymbol{w}^\top \boldsymbol{P}_{\boldsymbol{x}^+}^* - \boldsymbol{w}^\top \hat{\boldsymbol{P}}_{\boldsymbol{x}^+} \right) - \left( \boldsymbol{w}^\top \boldsymbol{P}_{\boldsymbol{x}^-}^* - \boldsymbol{w}^\top \hat{\boldsymbol{P}}_{\boldsymbol{x}^-} \right) \right|$$

$$= \xi_n^{-1} \sup_{\boldsymbol{w} \in \mathcal{W}} \left| \left\{ \sum_{m=1}^M w_m (\boldsymbol{\theta}_{(m)}^* - \hat{\boldsymbol{\theta}}_{(m)})^\top \frac{\partial \hat{P}_{(m),\boldsymbol{x}^+}}{\partial \hat{\boldsymbol{\theta}}_{(m)}} |_{\hat{\boldsymbol{\theta}}_{(m)} = \boldsymbol{\theta}_{(m),\boldsymbol{x}^+}^\star} \right\} - \left\{ \sum_{m=1}^M w_m (\boldsymbol{\theta}_{(m)}^* - \hat{\boldsymbol{\theta}}_{(m)})^\top \frac{\partial \hat{P}_{(m),\boldsymbol{x}^-}}{\partial \hat{\boldsymbol{\theta}}_{(m)}} |_{\hat{\boldsymbol{\theta}}_{(m)} = \boldsymbol{\theta}_{(m),\boldsymbol{x}^-}^\star} \right\} \right|$$

$$\leq C \xi_n^{-1} M O_p(n^{-1/2} M^{1/2})$$

$$= O_p(C \xi_n^{-1} n^{-1/2} M^{3/2})$$

$$= o_p(1),$$

$$(\text{A.4})$$

where $\boldsymbol{\theta}_{(m),\boldsymbol{x}^+}^\star$ and $\boldsymbol{\theta}_{(m),\boldsymbol{x}^-}^\star$ are both within $\mathcal{O}(\boldsymbol{\theta}_{(m)}^*, \varrho)$ defined in Assumption 2. The first inequality uses the fact: $\phi(x) = \ln(1 + e^{-x})$,

$$|\phi(x_1) - \phi(x_2)| = |\ln(1 + e^{-x_1}) - \ln(1 + e^{-x_2})| \leq |x_1 - x_2|, \qquad (\text{A.5})$$

the second equality uses Assumption 2, the second inequality uses Assumptions 1 and 2, and the last equality uses Assumption 4.

From this, we get:

$$\sup_{\boldsymbol{w} \in \mathcal{W}} \frac{\left| R_\phi(\boldsymbol{w}) - R_\phi^*(\boldsymbol{w}) \right|}{R_\phi^*(\boldsymbol{w})}$$

$$\leq \xi_n^{-1} \sup_{\boldsymbol{w} \in \mathcal{W}} \left| R_\phi(\boldsymbol{w}) - R_\phi^*(\boldsymbol{w}) \right|$$

$$= \xi_n^{-1} \sup_{\boldsymbol{w} \in \mathcal{W}} \left| \mathbb{E}_{\boldsymbol{x}^+ \sim \mathcal{P}^+, \boldsymbol{x}^- \sim \mathcal{P}^-} [\phi(\hat{P}_{\boldsymbol{x}^+}(\boldsymbol{w}) - \hat{P}_{\boldsymbol{x}^-}(\boldsymbol{w})) - \phi(P_{\boldsymbol{x}^+}^*(\boldsymbol{w}) - P_{\boldsymbol{x}^-}^*(\boldsymbol{w}))] \right|$$

$$\leq \mathbb{E}_{\boldsymbol{x}^+ \sim \mathcal{P}^+, \boldsymbol{x}^- \sim \mathcal{P}^-} \left( \xi_n^{-1} \sup_{\boldsymbol{w} \in \mathcal{W}} |\phi(\hat{P}_{\boldsymbol{x}^+}(\boldsymbol{w}) - \hat{P}_{x^-}(\boldsymbol{w})) - \phi(P_{x^+}^*(\boldsymbol{w}) - P_{x^-}^*(\boldsymbol{w}))| \right)$$

$$= o(1),$$

The second inequality uses Assumption 5, and the last step is due to Eq. A.4 and Assumption 5. Eq. A.1 is proven.

Next, we prove that Eq. A.3 holds. Note that:

$$\sup_{\boldsymbol{w} \in \mathcal{W}} \frac{\left| CV_\phi(\boldsymbol{w}) - R_\phi^*(\boldsymbol{w}) \right|}{R_\phi^*(\boldsymbol{w})}$$

$$\leq \left( \sup_{\boldsymbol{w} \in \mathcal{W}} \frac{\left| CV_\phi(\boldsymbol{w}) - CV_\phi^*(\boldsymbol{w}) \right|}{R_\phi^*(\boldsymbol{w})} + \sup_{\boldsymbol{w} \in \mathcal{W}} \frac{\left| CV_\phi^*(\boldsymbol{w}) - R_\phi^*(\boldsymbol{w}) \right|}{R_\phi^*(\boldsymbol{w})} \right). \qquad (\text{A.6})$$

For the first part of Eq. A.6,

$$\sup_{\boldsymbol{w}\in\mathcal{W}} \frac{\left|CV_\phi(\boldsymbol{w}) - CV_\phi^*(\boldsymbol{w})\right|}{R_\phi^*(\boldsymbol{w})}$$

$$\leq \xi_n^{-1} \sup_{\boldsymbol{w}\in\mathcal{W}} \left|CV_\phi(\boldsymbol{w}) - CV_\phi^*(\boldsymbol{w})\right|$$

$$= \xi_n^{-1} \sup_{\boldsymbol{w}\in\mathcal{W}} \left| \frac{1}{N_+N_-} \sum_{i=1}^{N_+} \sum_{j=1}^{N_-} \left[ \phi\left(\boldsymbol{w}^\top \tilde{\boldsymbol{P}}_{\boldsymbol{x}_i^+} - \boldsymbol{w}^\top \tilde{\boldsymbol{P}}_{\boldsymbol{x}_j^-}\right) - \phi\left(\boldsymbol{w}^\top \boldsymbol{P}_{\boldsymbol{x}_i^+}^* - \boldsymbol{w}^\top \boldsymbol{P}_{\boldsymbol{x}_j^-}^*\right) \right] \right|$$

$$\leq \xi_n^{-1} \sup_{\boldsymbol{w}\in\mathcal{W}} \frac{1}{N_+N_-} \sum_{i=1}^{N_+} \sum_{j=1}^{N_-} \left| \left(\boldsymbol{w}^\top \tilde{\boldsymbol{P}}_{\boldsymbol{x}_i^+} - \boldsymbol{w}^\top \tilde{\boldsymbol{P}}_{\boldsymbol{x}_j^-}\right) - \left(\boldsymbol{w}^\top \boldsymbol{P}_{\boldsymbol{x}_i^+}^* - \boldsymbol{w}^\top \boldsymbol{P}_{\boldsymbol{x}_j^-}^*\right) \right|$$

$$= \xi_n^{-1} \sup_{\boldsymbol{w}\in\mathcal{W}} \frac{1}{N_+N_-} \sum_{i=1}^{N_+} \sum_{j=1}^{N_-} \left| \left\{ \sum_{m=1}^M w_m \left(\boldsymbol{\theta}_{(m)}^* - \hat{\boldsymbol{\theta}}_{(m)}^{[-\sigma(i)]}\right)^\top \frac{\partial \tilde{P}_{(m),\boldsymbol{x}_i^+}^{[-\sigma(i)]}}{\partial \hat{\boldsymbol{\theta}}_{(m)}^{[-\sigma(i)]}} \big|_{\hat{\boldsymbol{\theta}}_{(m)}^{[-\sigma(i)]} = \boldsymbol{\theta}_{(m),i,+}^\star} \right\} \right.$$

$$\left. - \left\{ \sum_{m=1}^M w_m \left(\boldsymbol{\theta}_{(m)}^* - \hat{\boldsymbol{\theta}}_{(m)}^{[-\tau(j)]}\right)^\top \frac{\partial \tilde{P}_{(m),\boldsymbol{x}_j^-}^{[-\tau(j)]}}{\partial \hat{\boldsymbol{\theta}}_{(m)}^{[-\tau(j)]}} \big|_{\hat{\boldsymbol{\theta}}_{(m)}^{[-\tau(j)]} = \boldsymbol{\theta}_{(m),j,-}^\star} \right\} \right|$$

$$\leq C \xi_n^{-1} M O_p(n^{-1/2} M^{1/2})$$

$$= O_p(C \xi_n^{-1} n^{-1/2} M^{3/2})$$

$$= o_p(1),$$

where $\sigma(i)$ is a mapping indicating that $\boldsymbol{x}_i^+$ belongs to the $\sigma(i)$-th fold in the original $n$ observation samples, and $\tau(j)$ is similar. $\boldsymbol{\theta}_{(m),1,+}^\star, \ldots, \boldsymbol{\theta}_{(m),N_+,+}^\star$ and $\boldsymbol{\theta}_{(m),1,-}^\star, \ldots, \boldsymbol{\theta}_{(m),N_-,-}^\star$ are all within $\mathcal{O}(\boldsymbol{\theta}_{(m)}^*, \varrho)$ defined in Assumption 2. The second inequality is due to Eq. A.5, the second equality uses Assumption 2, the third inequality uses Assumptions 1 and 2, and the last equality uses Assumption 4.

For the second part of Eq. A.6, to prove $\sup_{\boldsymbol{w}\in\mathcal{W}} \frac{\left|CV_\phi^*(\boldsymbol{w}) - R_\phi^*(\boldsymbol{w})\right|}{R_\phi^*(\boldsymbol{w})} = o_p(1)$, according to Corollary 2.2 in Newey (1991), we only need to prove: (i) $\mathcal{W}$ is compact, (ii) for $\forall \boldsymbol{w} \in \mathcal{W}$, $\frac{CV_\phi^*(\boldsymbol{w}) - R_\phi^*(\boldsymbol{w})}{R_\phi^*(\boldsymbol{w})} = o_p(1)$, (iii) there exists $\kappa_T = O_p(1)$ such that for $\forall \boldsymbol{w}, \boldsymbol{w}' \in \mathcal{W}$, $|t(\boldsymbol{w}) - t(\boldsymbol{w}')| \leq \kappa_T \|\boldsymbol{w} - \boldsymbol{w}'\|$ holds, where $t(\boldsymbol{w}) = CV_\phi^*(\boldsymbol{w})/R_\phi^*(\boldsymbol{w}) - 1$.

(i) is obviously true, (iii) is Assumption 6, so we only need to prove (ii). Below we prove (ii). For $\forall \boldsymbol{w} \in \mathcal{W}$, for $\forall \delta > 0$:

$$\Pr\left( \left| \frac{CV_\phi^*(\boldsymbol{w}) - R_\phi^*(\boldsymbol{w})}{R_\phi^*(\boldsymbol{w})} \right| \geq \delta \right)$$

$$\leq \Pr\left( \left| CV_\phi^*(\boldsymbol{w}) - R_\phi^*(\boldsymbol{w}) \right| \geq \delta \xi_n \right)$$

$$= \Pr\left( \left| \frac{1}{N_+N_-} \sum_{i=1}^{N_+} \sum_{j=1}^{N_-} \phi\left(\boldsymbol{w}^\top \boldsymbol{P}_{\boldsymbol{x}_i^+}^* - \boldsymbol{w}^\top \boldsymbol{P}_{\boldsymbol{x}_j^-}^*\right) - \mathbb{E}_{\boldsymbol{x}^+ \sim \mathcal{P}^+, \boldsymbol{x}^- \sim \mathcal{P}^-} \left[ \phi\left(\boldsymbol{w}^\top \boldsymbol{P}_{\boldsymbol{x}^+}^* - \boldsymbol{w}^\top \boldsymbol{P}_{\boldsymbol{x}^-}^*\right) \right] \right| \geq \delta \xi_n \right)$$

$$\leq 2 e^{-\frac{2v\delta^2 \xi_n^2}{(C+1)^2 M^2}}$$

$$= o(1),$$

where the first inequality is because $\left| \frac{CV_\phi^*(\boldsymbol{w}) - R_\phi^*(\boldsymbol{w})}{R_\phi^*(\boldsymbol{w})} \right| \leq \xi_n^{-1} \left| CV_\phi^*(\boldsymbol{w}) - R_\phi^*(\boldsymbol{w}) \right|$. The second inequality uses Lemma 2 and the fact: $0 < \phi\left(\boldsymbol{w}^\top \boldsymbol{P}_{\boldsymbol{x}_i^+}^* - \boldsymbol{w}^\top \boldsymbol{P}_{\boldsymbol{x}_j^-}^*\right) < (C+1)M$ and $v = \min(N_+, N_-)$. The last equality is due to Assumptions 3 and 4.

This proves $\sup_{\boldsymbol{w}\in\mathcal{W}}\frac{|CV_\phi^*(\boldsymbol{w})-R_\phi^*(\boldsymbol{w})|}{R_\phi^*(\boldsymbol{w})} = o_p(1)$. So far, Eq. A.3 is proven, and the proof of Theorem 1 is completed.

## A.3 Proof of Theorem 2

Note that $\xi_n^* = \inf_{\boldsymbol{w}\in\mathcal{W}} R^*(\boldsymbol{w})$, and:

$$\xi_n^{*-1}\left( R(\hat{\boldsymbol{w}}) - \inf_{\boldsymbol{w}\in\mathcal{W}} R(\boldsymbol{w}) \right) = \xi_n^{*-1}\left( R(\hat{\boldsymbol{w}}) - R^*(\hat{\boldsymbol{w}}) \right) + \xi_n^{*-1}\left( R^*(\hat{\boldsymbol{w}}) - \inf_{\boldsymbol{w}\in\mathcal{W}} R^*(\boldsymbol{w}) \right)$$
$$+ \xi_n^{*-1}\left( \inf_{\boldsymbol{w}\in\mathcal{W}} R^*(\boldsymbol{w}) - \inf_{\boldsymbol{w}\in\mathcal{W}} R(\boldsymbol{w}) \right),$$

If we can prove:

$$\xi_n^{*-1}\left( R(\hat{\boldsymbol{w}}) - R^*(\hat{\boldsymbol{w}}) \right) = o_p(1), \tag{A.7}$$

$$\xi_n^{*-1}\left( R^*(\hat{\boldsymbol{w}}) - \inf_{\boldsymbol{w}\in\mathcal{W}} R^*(\boldsymbol{w}) \right) = o_p(1), \tag{A.8}$$

$$\xi_n^{*-1}\left( \inf_{\boldsymbol{w}\in\mathcal{W}} R^*(\boldsymbol{w}) - \inf_{\boldsymbol{w}\in\mathcal{W}} R(\boldsymbol{w}) \right) = o(1), \tag{A.9}$$

then we can get:

$$\xi_n^{*-1}\left( R(\hat{\boldsymbol{w}}) - \inf_{\boldsymbol{w}\in\mathcal{W}} R(\boldsymbol{w}) \right) = o_p(1), \quad \xi_n^{*-1}\inf_{\boldsymbol{w}\in\mathcal{W}} R(\boldsymbol{w}) = 1 + o(1),$$

From this, Theorem 2 follows immediately:

$$\frac{R(\hat{\boldsymbol{w}})}{\inf_{\boldsymbol{w}\in\mathcal{W}} R(\boldsymbol{w})} \xrightarrow{p} 1$$

First, we prove:

$$\xi_n^{*-1}\sup_{\boldsymbol{w}\in\mathcal{W}} |R(\boldsymbol{w}) - R^*(\boldsymbol{w})| = o(1), \tag{A.10}$$

If Eq. A.10 is proven, then Eq. A.7 and Eq. A.9 follow directly. Eq. A.10 is proven as follows:

$$\xi_n^{*-1}\sup_{\boldsymbol{w}\in\mathcal{W}} |R(\boldsymbol{w}) - R^*(\boldsymbol{w})|$$

$$= \xi_n^{*-1}\sup_{\boldsymbol{w}\in\mathcal{W}}\left| \mathbb{E}_{\boldsymbol{x}^+\sim\mathcal{P}^+,\boldsymbol{x}^-\sim\mathcal{P}^-}[\mathbb{I}\{\hat{P}_{\boldsymbol{x}^+}(\boldsymbol{w}) - \hat{P}_{\boldsymbol{x}^-}(\boldsymbol{w}) < 0\}] - \mathbb{E}_{\boldsymbol{x}^+\sim\mathcal{P}^+,\boldsymbol{x}^-\sim\mathcal{P}^-}[\mathbb{I}\{P_{\boldsymbol{x}^+}^*(\boldsymbol{w}) - P_{\boldsymbol{x}^-}^*(\boldsymbol{w}) < 0\}]\right|$$

$$= \frac{1}{p(1-p)}\xi_n^{*-1}\sup_{\boldsymbol{w}\in\mathcal{W}}\left| \mathbb{E}_{\boldsymbol{x}\sim\mathcal{P},\boldsymbol{x}'\sim\mathcal{P}}[\mathbb{I}\{\hat{P}_{\boldsymbol{x}}(\boldsymbol{w}) < \hat{P}_{\boldsymbol{x}'}(\boldsymbol{w})\}\mathbb{I}(y > y')] - \mathbb{E}_{\boldsymbol{x}\sim\mathcal{P},\boldsymbol{x}'\sim\mathcal{P}}[\mathbb{I}\{P_{\boldsymbol{x}}^*(\boldsymbol{w}) < P_{\boldsymbol{x}'}^*(\boldsymbol{w})\}\mathbb{I}(y > y')]\right|$$

$$= \frac{1}{2p(1-p)}\xi_n^{*-1}\sup_{\boldsymbol{w}\in W}\left| \mathbb{E}_{\boldsymbol{x}\sim\mathcal{P},\boldsymbol{x}'\sim\mathcal{P}}[\mathbb{I}\{(\hat{P}_{\boldsymbol{x}}(\boldsymbol{w}) - \hat{P}_{\boldsymbol{x}'}(\boldsymbol{w}))(y - y') < 0\}] - \mathbb{E}_{\boldsymbol{x}\sim\mathcal{P},\boldsymbol{x}'\sim\mathcal{P}}[\mathbb{I}\{(P_{\boldsymbol{x}}^*(\boldsymbol{w}) - P_{\boldsymbol{x}'}^*(\boldsymbol{w}))(y - y') < 0\}]\right|$$

$$= \frac{1}{2p(1-p)}\xi_n^{*-1}\sup_{\boldsymbol{w}\in\mathcal{W}}\left| \Pr\left[ (y - y') \times \sum_{m=1}^M w_m(\hat{P}_{(m),\boldsymbol{x}} - \hat{P}_{(m),\boldsymbol{x}'}) < 0 \right] - \Pr\left[ (y - y') \times \sum_{m=1}^M w_m(P_{(m),\boldsymbol{x}}^* - P_{(m),\boldsymbol{x}'}^*) < 0 \right]\right|$$

$$= \frac{1}{2p(1-p)}\xi_n^{*-1}\sup_{\boldsymbol{w}\in\mathcal{W}} |\Pr(a(\boldsymbol{w}) + \Delta(\boldsymbol{w}) < 0) - \Pr(a(\boldsymbol{w}) < 0)|$$

$$= \frac{1}{2p(1-p)}\xi_n^{*-1}\sup_{\boldsymbol{w}\in\mathcal{W}} \left| E_{\Delta(\boldsymbol{w})}[P(a(\boldsymbol{w}) + \Delta(\boldsymbol{w}) < 0|\Delta(\boldsymbol{w})) - P(a(\boldsymbol{w}) < 0|\Delta(\boldsymbol{w}))] \right|$$

$$\leq \frac{1}{2p(1-p)}E_{\Delta(\boldsymbol{w})}\left( \xi_n^{*-1}\sup_{\boldsymbol{w}\in\mathcal{W}} |[P(a(\boldsymbol{w}) + \Delta(\boldsymbol{w}) < 0|\Delta(\boldsymbol{w})) - P(a(\boldsymbol{w}) < 0|\Delta(\boldsymbol{w}))]| \right),$$
$$\tag{A.11}$$

where the second equality uses the fact:

$$\mathbb{E}_{\boldsymbol{x}\sim\mathcal{P},\boldsymbol{x}'\sim\mathcal{P}}[\mathbb{I}\{\hat{P}_{\boldsymbol{x}}(\boldsymbol{w}) < \hat{P}_{\boldsymbol{x}'}(\boldsymbol{w})\}\mathbb{I}(y > y')] = p(1-p)\mathbb{E}_{\boldsymbol{x}^+\sim\mathcal{P}^+,\boldsymbol{x}^-\sim\mathcal{P}^-}[\mathbb{I}\{\hat{P}_{\boldsymbol{x}^+}(\boldsymbol{w}) - \hat{P}_{\boldsymbol{x}^-}(\boldsymbol{w}) < 0\}],$$

the third equality uses the fact:

$$\mathbb{E}_{\boldsymbol{x}\sim\mathcal{P},\boldsymbol{x}'\sim\mathcal{P}}[\mathbb{I}\{(\hat{P}_{\boldsymbol{x}}(\boldsymbol{w}) - \hat{P}_{\boldsymbol{x}'}(\boldsymbol{w}))(y - y') < 0\}] = 2\mathbb{E}_{\boldsymbol{x}\sim\mathcal{P},\boldsymbol{x}'\sim\mathcal{P}}[\mathbb{I}\{\hat{P}_{\boldsymbol{x}}(\boldsymbol{w}) < \hat{P}_{\boldsymbol{x}'}(\boldsymbol{w})\}\mathbb{I}(y > y')],$$

the sixth equality is the law of total probability, and the last inequality uses Assumption 7.

Furthermore, we have:

$$
\begin{aligned}
&\xi_n^{*-1} \sup_{\boldsymbol{w}\in\mathcal{W}} |[P(a(\boldsymbol{w}) + \Delta(\boldsymbol{w}) < 0|\Delta(\boldsymbol{w})) - P(a(\boldsymbol{w}) < 0|\Delta(\boldsymbol{w}))]| \\
=&\xi_n^{*-1} \sup_{\boldsymbol{w}\in\mathcal{W}} \left|(F_{a(\boldsymbol{w})|\Delta(\boldsymbol{w})}(-\Delta(\boldsymbol{w})) - F_{a(\boldsymbol{w})|\Delta(\boldsymbol{w})}(0))\right| \\
=&\xi_n^{*-1} \sup_{\boldsymbol{w}\in\mathcal{W}} \left|\Delta(\boldsymbol{w}) f_{a(\boldsymbol{w})|\Delta(\boldsymbol{w})}(\varsigma)\right| \\
=&O_p(CM^{3/2}n^{-1/2}\xi_n^{*-1}) \\
=&o_p(1),
\end{aligned}
\tag{A.12}
$$

where the second equality is the mean value theorem, $\varsigma$ is between $0$ and $-\Delta(\boldsymbol{w})$, the third equality uses Assumptions 1, 2, and 7, and the last equality uses Assumption 9.

Therefore, from Eq. A.11 and Eq. A.12, we have:

$$
\begin{aligned}
&\xi_n^{*-1} \sup_{\boldsymbol{w}\in\mathcal{W}} |R(\boldsymbol{w}) - R^*(\boldsymbol{w})| \\
\leq&\frac{1}{2p(1-p)} E_{\Delta(\boldsymbol{w})}\left(\xi_n^{*-1} \sup_{\boldsymbol{w}\in\mathcal{W}} |[P(a(\boldsymbol{w}) + \Delta(\boldsymbol{w}) < 0|\Delta(\boldsymbol{w})) - P(a(\boldsymbol{w}) < 0|\Delta(\boldsymbol{w}))]|\right) \\
=&o(1),
\end{aligned}
$$

where the last step uses Eq. A.12 and Assumption 7. Eq. A.10 is proven.

According to Eq. A.10, we can immediately obtain Eq. A.7 and Eq. A.9. The derivation of Eq. A.7 is straightforward. The derivation of Eq. A.9 is as follows:

$$\inf_{\boldsymbol{w}\in\mathcal{W}}(R^*(\boldsymbol{w}) - R(\boldsymbol{w})) \leq \inf_{\boldsymbol{w}\in\mathcal{W}} R^*(\boldsymbol{w}) - \inf_{\boldsymbol{w}\in\mathcal{W}} R(\boldsymbol{w}) \leq \sup_{\boldsymbol{w}\in\mathcal{W}}(R^*(\boldsymbol{w}) - R(\boldsymbol{w})).$$

Next, we prove that Eq. A.8 holds. According to Assumption 8:

$$\xi_n^{*-1}\left(R^*(\hat{\boldsymbol{w}}) - \inf_{\boldsymbol{w}\in\mathcal{W}} R^*(\boldsymbol{w})\right) \leq 2\xi_n^{*-1}\sqrt{R_\phi^*(\hat{\boldsymbol{w}}) - \inf_{\boldsymbol{w}\in\mathcal{W}} R_\phi^*(\boldsymbol{w})},$$

We only need to prove

$$R_\phi^*(\hat{\boldsymbol{w}}) - \inf_{\boldsymbol{w}\in\mathcal{W}} R_\phi^*(\boldsymbol{w}) = o_p(1), \tag{A.13}$$

to obtain Eq. A.8.

According to Theorem 1 and the already proven Eq. A.1, we get:

$$\xi_n^{-1}\left(R_\phi^*(\hat{\boldsymbol{w}}) - \inf_{\boldsymbol{w}\in\mathcal{W}} R_\phi^*(\boldsymbol{w})\right) = o_p(1),$$

From this, Eq. A.13 follows immediately. This completes the proof of Eq. A.8. So far, Eq. A.7, A.8, and A.9 are all proven, and Theorem 2 is proven.

### A.4 PROOF OF THEOREM 3

Theorem 3 can be divided into two parts: $\boldsymbol{w}^* = \arg\min_{\boldsymbol{w}\in\mathcal{W}} R^*(\boldsymbol{w})$ and $\hat{\boldsymbol{w}} \xrightarrow{p} \boldsymbol{w}^*$. We first prove $\hat{\boldsymbol{w}} \xrightarrow{p} \boldsymbol{w}^*$. According to Theorem 5.7 in Van der Vaart (2000), we only need to prove that the following holds:

$$\sup_{\boldsymbol{w}\in\mathcal{W}} |CV_\phi(\boldsymbol{w}) - R_\phi^*(\boldsymbol{w})| = o_p(1), \tag{A.14}$$

Combined with $\boldsymbol{w}^* = \arg\min_{\boldsymbol{w}\in\mathcal{W}} R_\phi^*(\boldsymbol{w})$ and the uniqueness of $\boldsymbol{w}^*$, we get $\hat{\boldsymbol{w}} \xrightarrow{p} \boldsymbol{w}^*$. The proof of Eq. A.14 is similar to that of Eq. A.3 and will not be repeated here.

Then we prove $\boldsymbol{w}^* = \underset{\boldsymbol{w} \in \mathcal{W}}{\arg\min} \ R^*(\boldsymbol{w})$. According to Assumption 8, we have: $R^*(\boldsymbol{w}^*) - \inf_{\boldsymbol{w} \in \mathcal{W}} R^*(\boldsymbol{w}) \leq 2\sqrt{R^*_\phi(\boldsymbol{w}^*) - \inf_{\boldsymbol{w} \in \mathcal{W}} R^*_\phi(\boldsymbol{w})}$. Combined with $\boldsymbol{w}^* = \underset{\boldsymbol{w} \in \mathcal{W}}{\arg\min} \ R^*_\phi(\boldsymbol{w})$, we immediately get $\boldsymbol{w}^* = \underset{\boldsymbol{w} \in \mathcal{W}}{\arg\min} \ R^*(\boldsymbol{w})$. The proof of Theorem 3 is complete.

### A.5 NUMERICAL SIMULATION

In this section, we evaluate the performance of our ELAM method in two experimental settings: (i) when correct model is a part of the base model space, and (ii) when correct model is not included in the base model space.

### A.5.1 SIMULATION SETTING 1: VERIFICATION OF WEIGHT CONSISTENCY

To empirically validate the weight consistency property established in Theorem 3, we adapt the simulation framework from Zhang & Liu (2023) to a binary classification setting. The data-generating process is specified as:

$$P(y_i = 1 | \boldsymbol{x}_i) = \frac{\exp(\eta_i)}{1 + \exp(\eta_i)},$$

where $\eta_i = \sum_{j=1}^{q} \beta_j x_{ji}$, $\boldsymbol{x}_i = (x_{1i}, \ldots, x_{qi})^\top \sim N(\boldsymbol{0}, \Omega)$, the diagonal elements of $\Omega$ are 1, and the off-diagonal elements are 0.5. The true model only has the first $p$ coefficients nonzero. We set $p = 4$, $q = 20$. The regression coefficient $\boldsymbol{\beta} = (1, 1, 1, c, 0, \ldots, 0)^\top$, where the parameter $c$ takes values in $\{0.01, 0.1, 0.2, ..., 0.99\}$.

We consider two base models: a correctly specified model including all $q$ features, and a misspecified model including the first $p - 1$ features. This design allows us to examine how ELAM's weight allocation responds to varying degrees of model misspecification, controlled by parameter $c$. The misspecified and correctly specified models are denoted as Incorrect and Correct, respectively. The larger the parameter $c$, the greater the impact of the missing covariate on the misspecified base model.

According to the scale-invariance of the AUC risk of the stacking estimator, let $\tilde{\boldsymbol{w}} = \frac{\hat{\boldsymbol{w}}}{\|\hat{\boldsymbol{w}}\|_1}$. If the weight $\tilde{\boldsymbol{w}}$ placed on the correct base model converges to 1 as the sample size and parameter $c$ increase, Theorem 3 is verified.

For each combination of sample size $n \in \{100, 500, 2000, 5000\}$ and parameter $c$, we repeatedly generate 200 independent datasets, employing a 70%-30% train-test split. Performance is evaluated via relative empirical AUC risk, normalized against ELAM's performance, called relative empirical AUC risk. This metric less than 1 indicates that the method's predictive performance is better than ELAM.

Figure 2 shows that when the sample size $n = 100$, the empirical AUC risk of the correct base model is significantly higher than that of the incorrect base model. When the sample size increases to $n = 500$ or 2000, for smaller parameter values $c$, the predictive performance of the correct base model is still worse than that of the incorrect base model, but the situation reverses as the parameter value $c$ increases. When the sample size $n = 5000$, the correct base model is almost consistently better than the incorrect base model. For the proposed ELAM method, when the sample size is 100, its predictive performance is close to that of the better incorrect base model. When the sample size is 500, the ELAM method is significantly better than both base models for larger parameter values $c$. When the sample size is 2000 or 5000, the ELAM method always tends to perform as well as, or even better than, the best base model.

Figure 3 provides direct evidence for Theorem 3's weight consistency guarantee. ELAM's weight allocation to the correctly specified model increases monotonically with both sample size and misspecification severity $c$, approaching 0.83 for $n = 5000$ and $c = 0.99$. This empirical validation confirms that ELAM successfully identifies and leverages correctly specified models when sufficient data is available.

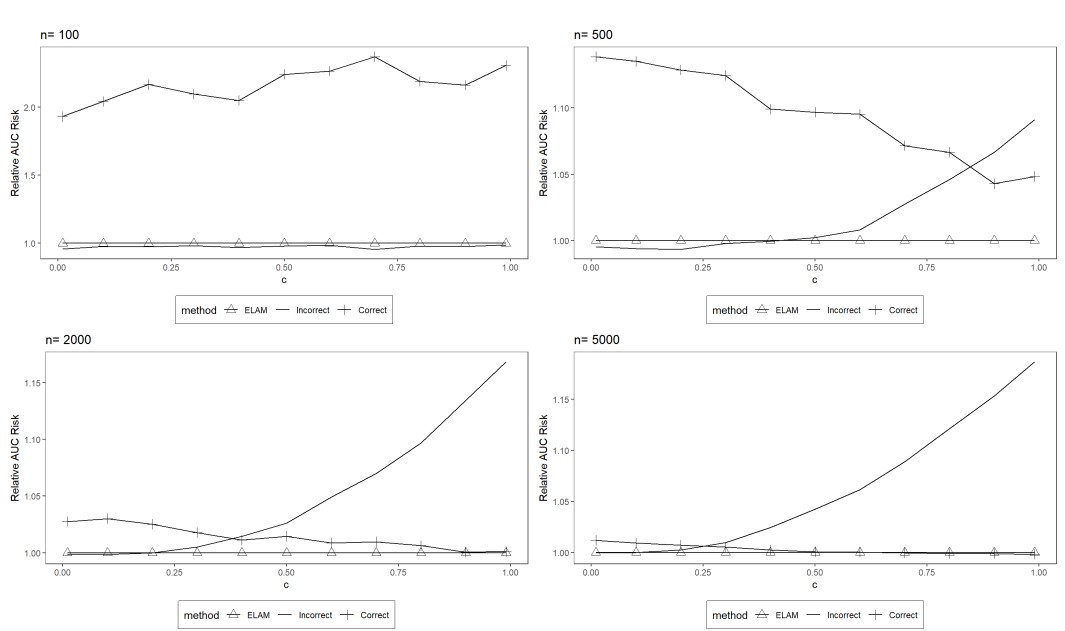

Figure 2: Relative empirical AUC risk of methods in Simulation Setting 1.

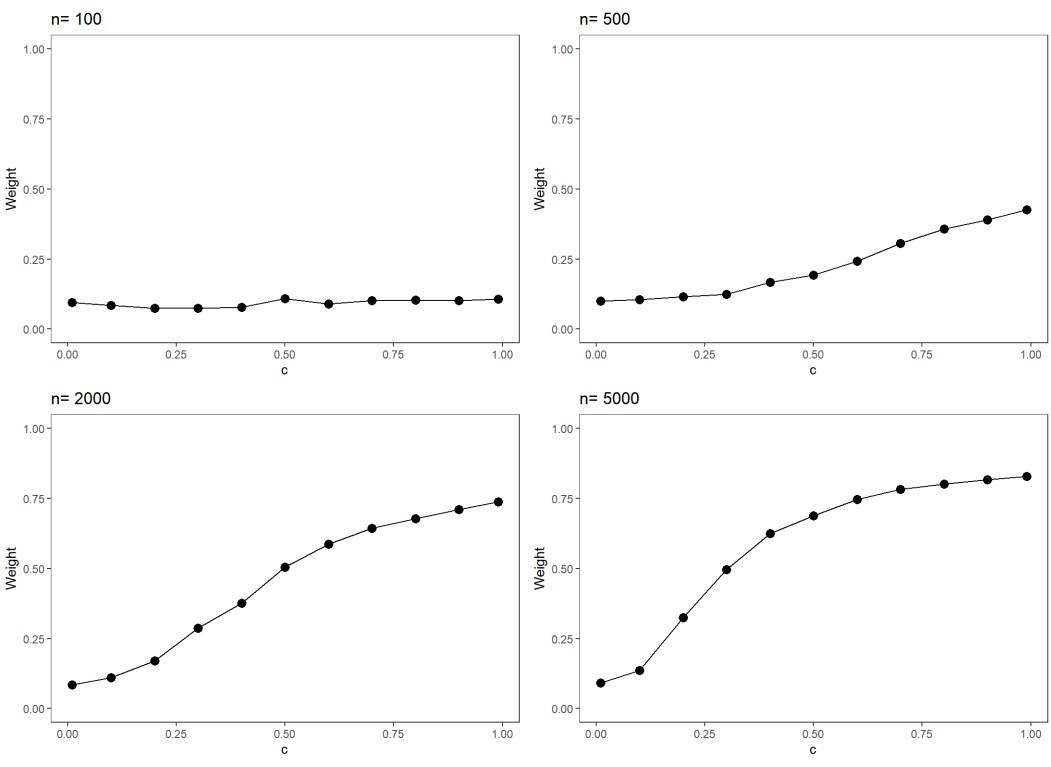

Figure 3: Average normalized weight placed on the correct base model in Simulation Setting 1.

A.5.2   SIMULATION SETTING 2: ROBUSTNESS UNDER MODEL MISSPECIFICATION

We further examine ELAM's performance under a more challenging misspecification scenario, employing the data-generating process:

$$P(y_i = 1|\boldsymbol{x}_i) = \frac{I(\boldsymbol{x}_i^\top \boldsymbol{\beta}_0 > -\frac{1}{4})}{1 + (1 + 4\boldsymbol{x}_i^\top \boldsymbol{\beta}_0)^{-\frac{1}{4}}},$$

where $\boldsymbol{x}_i = (x_{i1}, x_{i2}, ..., x_{ip})^\top$, $x_{ik} = z_{ik}^2 - 1$, $k = 1, 2, \ldots, p$. $\boldsymbol{z}_i = (z_{i1}, z_{i2}, \ldots, z_{ip})^\top$ is a $p$-dimensional multivariate normal vector with mean 0 and covariance matrix $\Sigma = (\Sigma_{i,j}) = (0.5^{|i-j|})$. Fix $p = 9$. The regression coefficient $\boldsymbol{\beta}_0 = (1, 1, 0.1, 0.1, 0, 0, 0, c)^\top$, and we let the parameter $c$ vary over $\{0.01, 0.1, 0.2, ..., 0.99\}$. We assume the last dimension covariate is unobserved. We consider nested (8 models) and non-nested ($2^8 - 1$ models) cases, both with logistic regression base learners.

For each parameter $c$, we randomly generate 1000 samples from the aforementioned distribution. As in Simulation Setting 1, randomly split the sample into training and testing sets in a 7:3 ratio, and calculate the relative empirical AUC risk of each method through 200 repeated simulation experiments. Tables 1 and 2 show the results for nested and non-nested base models, respectively. In both cases, ELAM demonstrates robust performance across both nested and non-nested settings, achieving best or second-best performance in 19 of 22 configurations. The advantage is particularly pronounced in the non-nested case, where ELAM outperforms the ME method by up to 25.7%. These results highlight ELAM's ability to effectively navigate complex model spaces while maintaining computational tractability.

Table 1   Simulation Setting 2, Nested Base Models

| ELAM | ME | AUCW | SA | FULL | SAIC | SBIC | AIC | BIC | c |
|---|---|---|---|---|---|---|---|---|---|
| 1.0000 | **0.9936** | 1.0057 | 1.0068 | 1.0805 | 1.0176 | 0.9968 | 1.0337 | 1.0098 | 0.01 |
| 1.0000 | **0.9940** | 1.0032 | 1.0039 | 1.0753 | 1.0137 | 0.9993 | 1.0286 | 1.0108 | 0.1 |
| 1.0000 | **0.9932** | 1.0006 | 1.0017 | 1.0609 | 1.0129 | 1.0019 | 1.0298 | 1.0159 | 0.2 |
| 1.0000 | **0.9962** | 1.0015 | 1.0015 | 1.0591 | 1.0169 | 1.0016 | 1.0335 | 1.0112 | 0.3 |
| **1.0000** | 1.0010 | 1.0036 | 1.0035 | 1.0513 | 1.0196 | 1.0063 | 1.0361 | 1.0123 | 0.4 |
| 1.0000 | 1.0023 | **0.9977** | 0.9978 | 1.0308 | 1.0158 | 1.0035 | 1.0334 | 1.0102 | 0.5 |
| **1.0000** | 1.0063 | 1.0047 | 1.0052 | 1.0225 | 1.0162 | 1.0119 | 1.0345 | 1.0162 | 0.6 |
| **1.0000** | 1.0010 | 1.0060 | 1.0064 | 1.0290 | 1.0160 | 1.0133 | 1.0316 | 1.0170 | 0.7 |
| **1.0000** | 1.0077 | 1.0055 | 1.0056 | 1.0252 | 1.0158 | 1.0158 | 1.0296 | 1.0212 | 0.8 |
| **1.0000** | 1.0059 | 1.0105 | 1.0108 | 1.0115 | 1.0112 | 1.0236 | 1.0242 | 1.0282 | 0.9 |
| **1.0000** | 1.0094 | 1.0098 | 1.0104 | 1.0119 | 1.0142 | 1.0207 | 1.0269 | 1.0257 | 0.99 |

**Bold numbers** and underlined numbers indicate the best and second best items, respectively

Table 2   Simulation Setting 2, Non-nested Base Models

| ELAM | ME | AUCW | SA | FULL | SAIC | SBIC | AIC | BIC | c |
|---|---|---|---|---|---|---|---|---|---|
| 1.0000 | 1.2574 | 1.0239 | 1.0423 | 1.0688 | 1.0147 | **0.9887** | 1.0419 | 1.0040 | 0.01 |
| 1.0000 | 1.2402 | 1.0179 | 1.0354 | 1.0642 | 1.0126 | **0.9876** | 1.0439 | 1.0051 | 0.1 |
| 1.0000 | 1.2105 | 1.0217 | 1.0389 | 1.0559 | 1.0091 | **0.9895** | 1.0339 | 1.0082 | 0.2 |
| 1.0000 | 1.2257 | 1.0240 | 1.0398 | 1.0546 | 1.0086 | **0.9875** | 1.0371 | 1.0058 | 0.3 |
| 1.0000 | 1.1659 | 1.0159 | 1.0283 | 1.0466 | 1.0087 | **0.9934** | 1.0332 | 1.0161 | 0.4 |
| **1.0000** | 1.1439 | 1.0091 | 1.0198 | 1.0312 | 1.0082 | 1.0034 | 1.0288 | 1.0259 | 0.5 |
| **1.0000** | 1.1375 | 1.0097 | 1.0188 | 1.0349 | 1.0054 | 1.0060 | 1.0240 | 1.0188 | 0.6 |
| **1.0000** | 1.1071 | 1.0107 | 1.0188 | 1.0308 | 1.0046 | 1.0031 | 1.0243 | 1.0152 | 0.7 |
| **1.0000** | 1.1316 | 1.0056 | 1.0115 | 1.0233 | 1.0055 | 1.0070 | 1.0188 | 1.0261 | 0.8 |
| **1.0000** | 1.1021 | 1.0070 | 1.0134 | 1.0302 | 1.0071 | 1.0007 | 1.0211 | 1.0205 | 0.9 |
| **1.0000** | 1.0954 | 1.0093 | 1.0147 | 1.0232 | 1.0024 | 1.0048 | 1.0146 | 1.0198 | 0.99 |

**Bold numbers** and underlined numbers indicate the best and second best items, respectively

## A.6 Extra Experiments

In this section, we evaluate the performance of ELAM on two extra real datasets: the MONK's Problems Dataset Wnek (1993) and the Credit Approval Dataset Quinlan (1987) from the UCI Machine Learning Repository.

The MONK's Problems comprises 432 observations, each with 6 features. The Credit Approval Dataset contains 653 credit card application records (from an original 690 instances after preprocessing), each characterized by 15 financial and demographic attributes.

Follow the base model set in section 5, for the MONK's Problems Dataset, we consider the complete non-nested model space consisting of $2^6 - 1 = 63$ distinct feature combinations. For the Credit Approval Dataset, we restrict to a nested sequence of 15 models obtained by sequentially adding features based on their original order. In both cases, we employ logistic regression as the base learner.

We randomly split the sample into training and testing sets in a 7:3 ratio, and calculate the relative empirical AUC risk of each method through 200 repeated experiments. Figure 4 presents the empirical AUC risk distributions across all methods on both datasets. ELAM demonstrates consistent superiority, achieving the lowest AUC risk in both experimental settings.

On the MONK's Problems Dataset, ELAM reduces average AUC risk by substantial margins compared to all competitors: **34.39%** versus ME, **3.66%** versus AUCW, **3.85%** versus SA, **7.13%** versus FULL, **4.22%** versus SAIC, **2.87%** versus SBIC, **5.41%** versus AIC, **5.96%** versus BIC.

The performance advantage remains pronounced on the Spambase Dataset, with relative risk reductions of **3.73%** (ME), **9.96%** (AUCW), **13.43%** (SA), **2.76%** (FULL), **1.75%** (SAIC), **8.04%** (SBIC), **2.76%** (AIC), **10.96%** (BIC).

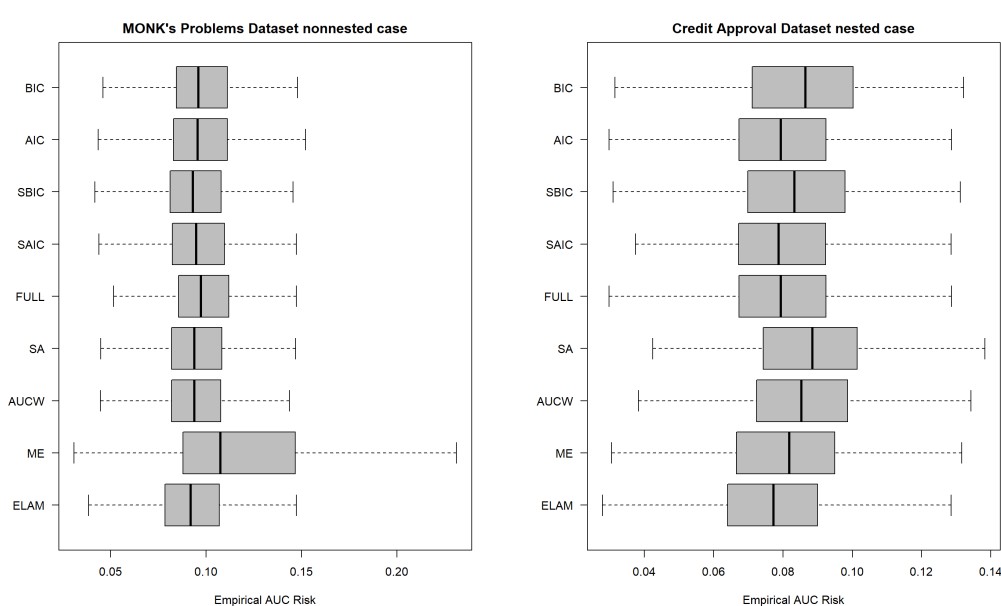

Figure 4: Empirical AUC risk of models on the MONK's Problems Dataset and Spambase Dataset.

