# OpenReview forum: "Ensemble Learning for AUC Maximization via Surrogate Loss"
_ICLR.cc/2026/Conference — ICLR 2026 Conference Withdrawn Submission_

### Official Review · Reviewer_zy6d · 2025-10-27

**Soundness:** 2
**Presentation:** 2
**Contribution:** 1
**Rating:** 2
**Confidence:** 3

**Summary:**

In this paper, the authors proposed an optimization framework named Ensemble Learning method for AUC Maximization **(ELAM)**. To be specific, ELAM aims to learn the optimal weight (mixture weight) of an ensemble of models so that the "surrogate" AUC is maximized. Here, surrogate means replacing the indicator function in the definition of AUC with some continuous approximations such as quadratic, exponential, or logistic functions. The base models in the ensemble are learned in a cross-validated manner.

In terms of theory, under technical assumptions, the authors show the consistency of the learned mixture weight in terms of:

- Theorem 1: Asymptotically optimal with respect to the surrogate risk objective.
- Theorem 2: Asymptotically optimal with respect to the original AUC.
- Theorem 3: Convergence with probability one to the optimal solution.

Numerical experiments show improved AUC compared to baselines like BIC, AIC, or other stacking methods such as AUCW and simple averaging.

**Strengths:**

- Directly maximizing AUC is meaningful and practically relevant, particularly in scenarios where the data are imbalanced.
- The paper is generally well written.

**Weaknesses:**

- The paper has limited novelty. The idea of replacing the indicator function by smoother ones should be very standard.

**Questions:**

- The authors used a handful number of assumptions: 5 assumptions to derive Theorem 1; 9 assumptions to derive Theorem 2; and 5 assumptions to derive Theorem 3. Are these assumptions easy to verify in practice?

- I wonder how the method stands against other methods outside the ensemble family, i.e., those that maximize the surrogate AUC and are not necessarily given as a weighted sum of pretrained models.

- The authors mentioned that Assumption 4 is stronger than those in the literature because they allow weights to be upper-bounded by $C$ instead of 1. In the case where the upper bound is exactly $1$, can Assumption 4 be alleviated to milder assumptions as in the literature?

---

### Official Review · Reviewer_V8g7 · 2025-10-30

**Soundness:** 3
**Presentation:** 2
**Contribution:** 2
**Rating:** 4
**Confidence:** 4

**Summary:**

This paper proposes an ensemble method named ELAM, designed to directly optimize the AUC via a surrogate loss. The core idea is to use K-fold cross-validation to generate out-of-sample predictions for each base learner and then learn an optimal linear combination by minimizing a differentiable pairwise surrogate loss (e.g., logistic loss) under box constraints. Experiments show that this method has certain effectiveness.

**Strengths:**

1. The approach is straightforward: generate out-of-sample predictions via cross-validation and learn combination weights by minimizing a pairwise surrogate loss. It can be easily applied to any collection of base models without architecture modification.
2. The paper presents a comprehensive asymptotic analysis, proving surrogate risk and AUC risk optimality as well as consistency of the learned weights.

**Weaknesses:**

1. The theoretical results rely on several restrictive and technical assumptions (e.g., bounded gradients, specific convergence rates, and conditions on $\xi_n$ and $M$). These assumptions may not hold in high-dimensional or deep-learning settings. The paper would benefit from intuitive explanations of these assumptions and examples of when they are satisfied or violated.
2. Experiments are conducted only on small-to-medium UCI datasets with logistic regression as base learners. No comparison is made with modern deep AUC optimization methods or large-scale ensemble approaches.
3. The robustness of ELAM with respect to class imbalance or hyperparameter variation is unclear.

**Questions:**

1. Several assumptions (e.g., Assumptions 4 and 9) restrict the growth of M relative to n. Could the authors provide intuition or numerical examples showing these are realistic in practice?
2. Could the authors compare ELAM with recent deep AUC optimization methods [1,2,3,4]?

[1] LibAUC: A Deep Learning Library for X-risk Optimization

[2] Algorithmic Foundation of Empirical X-risk Minimization

[3] Learning with Multiclass AUC: Theory and Algorithms

[4] AUCSeg: AUC-oriented Pixel-level Long-tail Semantic Segmentation

---

### Official Review · Reviewer_8GWi · 2025-10-31

**Soundness:** 3
**Presentation:** 3
**Contribution:** 2
**Rating:** 6
**Confidence:** 3

**Summary:**

The paper proposes ELAM (Ensemble Learning for AUC Maximization), a stacking-based ensemble method that directly targets AUC maximization through a surrogate loss.
Direct optimization of AUC over ensemble weights is NP-hard, so the authors propose to replace the non-differentiable AUC indicator with a smooth, convex surrogate trained using K-fold cross-validation to obtain unbiased out-of-sample predictions from base learners.
Theoretical results establish asymptotic optimality under both the surrogate risk and the true AUC risk, as well as weight consistency, showing that the method asymptotically concentrates weights on correctly specified base models.
Empirical results on real-world and synthetic datasets suggest that ELAM achieves modest but consistent improvements in AUC compared to existing ensemble and model averaging baselines.

**Strengths:**

The paper is generally well-written and clearly structured. The problem is well-motivated, and the proposed method (ELAM) is technically sound, though largely builds on existing surrogate loss ideas for AUC optimization. The theoretical analysis is thorough, with detailed proofs that support the main claims. Overall, the presentation is clear, and the technical development is sounds.

**Weaknesses:**

- Limited Novelty.
Replacing the AUC indicator with a convex surrogate and training weights via cross-validation is well-known in AUC optimization (Gao & Zhou 2015; LeDell et al. 2016). The contribution largely builds on these ideas within a stacking setup, adding theoretical justification but limited methodological novelty.
- Restrictive Setting and Scalability Concerns.
The method is essentially a linear stacking with logistic surrogate loss; it scales poorly with the number of base models and samples due to quadratic pairwise terms.

**Questions:**

How does the proposed surrogate differ fundamentally from prior logistic/exponential AUC-consistent surrogates used in ranking optimization?
What is the computational complexity of optimizing the pairwise surrogate (Eq. 11) in terms of n and M? Can it scale to large datasets?
How sensitive are results to the choice of surrogate loss (logistic vs. exponential) and the number of folds K in cross-validation?

---

### Official Review · Reviewer_KsJN · 2025-11-01

**Soundness:** 1
**Presentation:** 3
**Contribution:** 1
**Rating:** 0
**Confidence:** 3

**Summary:**

This paper proposes ELAM, an ensemble learning framework for maximizing the AUC for binary classification problems by combining base classifiers through a stacking approach. ELAM formulates the ensemble weighting problem as minimizing a surrogate pairwise loss based on K-fold cross-validation predictions to approximate direct AUC optimization. The authors provide theoretical guarantees demonstrating asymptotic optimality of the ensemble in surrogate and true AUC risk.

**Strengths:**

Theoretical Guarantees: The paper rigorously proves that the stacking weights computed by minimizing the surrogate loss converge asymptotically to the optimal solution in terms of both surrogate and actual AUC risk.

**Weaknesses:**

1. The novelty of the proposed method is questionable. A learning scheme is a standard stacking method with k-fold cross-validation. Authors should refer to previous literature in Section 3, e.g., “Using Stacking Approaches for Machine Learning Models” by Bohdan Pavlyshenko. A surrogate loss is adopted from (Gao & Zhou, 2015). So what is the novelty of the method?

2. In the introduction, authors state: “We address the NP-hard challenge of direct AUC optimization”. But no statement about the particular algorithmic problem and its NP-hardness do not follow after that.

3. line 161: R_{\phi} is not defined.

**Questions:**

1. When you say “the NP-hard nature of direct AUC optimization”, what do you mean specifically? When you make such a statement, I expect either a reference to previous literature or at least a particular formulation of an algorithmic problem that is NP-hard. Real-valued optimization problems in equations (5), (6) do not belong to a class of discrete problems which could be NP-hard or not NP-hard.

Note that exponentially complex ALGORITHMS and NP-hard PROBLEMS  are different concepts; please do not conflate them. There are previous papers with similar claims like “One-Pass AUC Optimization” by Gao et al., however, without any specification (“Direct optimization of AUC often leads to an NP-hard problem as it can be cast into a combinatorial optimization problem”). What is the formal input data for the problem you name NP-hard?

2. line 161: R_{\phi} is not defined.

---

### Note · Authors · 2025-11-21

I have read and agree with the venue's withdrawal policy on behalf of myself and my co-authors.